# Factuality Enhanced Language Models for Open-Ended Text Generation

Nayeon Lee[*†1], Wei Ping[†2], Peng Xu[2], Mostofa Patwary[2], Pascale Fung[1],
Mohammad Shoeybi[2], and Bryan Catanzaro[2]

[1]Hong Kong University of Science and Technology
[2]NVIDIA

## Abstract

Pretrained language models (LMs) are susceptible to generate text with nonfactual information. In this work, we measure and improve the factual accuracy of large-scale LMs for open-ended text generation. We design the FACTUALITYPROMPTS test set and metrics to measure the factuality of LM generations. Based on that, we study the factual accuracy of LMs with parameter sizes ranging from 126M to 530B. Interestingly, we find that larger LMs are more *factual* than smaller ones, although a previous study suggests that larger LMs can be less truthful in terms of *misconceptions*. In addition, popular sampling algorithms (e.g., top-$p$) in open-ended text generation can harm the factuality due to the "uniform randomness" introduced at every sampling step. We propose the *factual-nucleus* sampling algorithm that dynamically adapts the randomness to improve the factuality of generation while maintaining quality. Furthermore, we analyze the inefficiencies of the standard training method in learning correct associations between entities from factual text corpus (e.g., Wikipedia). We propose a *factuality-enhanced* training method that uses TOPICPREFIX for better awareness of facts and sentence completion as the training objective, which can vastly reduce the factual errors.

## 1 Introduction

Large-scale pre-trained language models (LMs) have demonstrated impressive natural language generation results [1–4]. However, the generative LMs (e.g., GPT-3) are solely trained to model the statistical correlations between subword tokens [5], and have limited capability to generate factually accurate text as illustrated in Table 1. As a result, there are increasing concerns about the nonfactual generations from large-scale pre-trained LMs [e.g., 6–8], which needs to be adequately addressed for their safe deployment in real-world applications, e.g., content creation [9] and dialogue [10].

In previous studies, different metrics and methods have been proposed to measure and improve the factual accuracy of language generation within different tasks [11], including text summarization [e.g., 12–15], question answering [e.g., 16–18], and table-to-text generation [e.g., 19, 20]. However, these works focus on the faithfulness (or factuality) of the *fine-tuned* LMs for particular downstream tasks (i.e., factual consistency between source and target text). Little exploration has been made to address the factual errors in *pretrained* LMs for general-purpose open-ended text generation, where the goal is to generate a coherent continuation from the given context (e.g., the use cases from GPT-2).

One of the popular methods for enhancing generation factuality is to incorporate external knowledge sources [21–23]. Structured knowledge bases and graphs have been utilized for grounded

---

[*]Work done during an internship at NVIDIA.
[†]Correspondence to: Nayeon Lee <nayeon.lee@connect.ust.hk>, Wei Ping <wping@nvidia.com>.

text generation [e.g., 24, 25], where the LMs are trained to select and copy relevant facts from external knowledge sources. In contrast to the sizeable online text with factual information, the structured knowledge graphs only encode a limited amount of knowledge as they require expensive human annotations for high-quality construction. A method that can directly leverage plain text knowledge (e.g., Wikipedia, encyclopedia books, peer-reviewed publications) would be desirable for factuality enhancement as it can remove the human annotation bottleneck and easily scale up the amount of injected knowledge. Augmenting LM with an information retrieval (IR) system is one possible solution to leverage textual facts, however, at the cost of additional complexity and resource overhead to the model [10, 26, 22, 27, 28]. Therefore, we explore an IR-free method that enhances the innate factuality of LMs by continued training on a factually rich plain-text corpus.

In this work, we focus on measuring and improving the factuality of large-scale pre-trained language models (LMs) for open-ended text generation. Specifically, we make the following contributions:

1. We build the benchmark and metrics [3] to measure the factual accuracy of pre-trained LM for open-ended text generation. We demonstrate a good correlation between the proposed automatic metrics and human assessment of factuality. Based on that, we systematically study the factual accuracy of LMs with parameter sizes ranging from 126M to 530B and find that large LMs have higher factual accuracy than smaller ones (e.g., named-entity factual error is reduced from 63.69% to 33.3%).

2. We study the decoding algorithms of LM in terms of factual accuracy. We unveil that the popular nucleus sampling algorithm [29] for open-ended text generation can easily mix up different named entities or randomly fabricate information due to the "uniform randomness" introduced at every decoding step. We propose *factual-nucleus* sampling algorithm that promotes generation factuality while maintaining the quality and diversity.

3. We explore training methods that can effectively leverage text corpus with rich facts (e.g., Wikipedia). We find that directly continuing the training of LM on factual text data [30] does not guarantee the improvement of factual accuracy. We propose *factuality-enhanced* training to address the underlying inefficiencies of this baseline. Our method consists of i) an addition of a TOPICPREFIX that improves the awareness of facts during training, and ii) a sentence completion task as the new objective for continued LM training [e.g., 30].

4. We demonstrate that the factual accuracy of large-scale LMs (up to 530B) can be significantly enhanced (i.e., named-entity factual error is reduced from 33.3% to 14.5%) after applying the proposed *factuality-enhanced* training with *factual-nucleus* sampling algorithm.

We organize the rest of the paper as follows. We discuss related work in § 2 and present our benchmark setup with evaluation protocol in § 3. We study the factual accuracy of LMs with respect to model size, prompt type, and choice of decoding algorithm in § 4. After that, we present *factual-nucleus* sampling algorithm in § 5, and *factuality-enhanced* training in § 6. We conclude the paper in § 7.

## 2 Related Work

**Factuality vs. Model Size** Lin et al. [31] propose the TruthfulQA benchmark to measure the falsehood generations from different sized LMs. The result suggests that bigger LMs pre-trained on web text are generally less truthful than smaller ones in terms of false belief or misconception. At first glance, this is contradictory to our observation, however, our work focuses on different knowledge to TruthfulQA work. The TruthfulQA benchmark focuses on conceptual knowledge, while our benchmark focuses on factual knowledge [32] [4]. Large LMs can be good at recalling factual knowledge given substantial pre-training corpus, suggested by previous studies on LM's parameteric knowledge [33], but there still remains room for improvement for reasoning conceptual knowledge [34, 35].

**Parametric Factual Knowledge** A group of work addresses the factual errors in the parametric knowledge of LMs that is acquired from training corpus [36–38]. The correctness of the parametric knowledge is commonly tested in cloze-style question answering format [33] (e.g., Person X is born in __). Efforts are made to fine-tune the pre-trained LM to "inject" more knowledge and improve its ability to answer factual questions without consulting external knowledge source [17].

---

[3]The implementation can be found in https://github.com/nayeon7lee/FactualityPrompt

[4]According to Krathwohl [32], knowledge can be categorized into four types: i) factual knowledge, ii) conceptual knowledge, iii) procedural knowledge, and iv) metacognitive knowledge.

Table 1: Example of continuations from the 530B LM with greedy decoding and nucleus sampling $p = 0.9$, along with the continuation from factuality-enhanced 530B LM with *factual-nucleus sampling*. Red represents nonfactual, green represents factual, and ~~strikethrough~~ represents repetition. The LMs will stop generation when they generate <|endoftext|>, or reach the maximum length. See Appendix A for more examples.

| |
|---|
| **[Factual Prompt]** One of lung cancer's symptoms is shortness of breath. |
| **[Greedy]** The American Cancer Society estimates that more than 228,000 people will be diagnosed with lung cancer in the United States this year.\n\n ~~The American Cancer Society estimates that more than 228,000 people will be diagnosed with lung cancer in the United States this year.\n\nThe American Cancer Society estimates that more than 228,000 people will be diagnosed with lung cancer in the United...~~ |
| **[p=0.9]** That would make an oxygen mask one of the more popular treatments for this devastating disease. It helps ease breathing and give patients back their strength. Learn more here.<|endoftext|>" |
| **[Ours]** Other symptoms include coughing (including coughing up blood), chest pain, hoarseness, weight loss, and loss of appetite. The symptoms of lung cancer may resemble those of a bad cold or pneumonia. A chest x - ray is necessary to confirm the diagnosis. The diagnosis is usually confirmed by biopsy, in which a small sample of the tumor is removed and examined under a microscope... |
| **[Non-Factual Prompt]** Cancer can be prevented by drinking too much alcohol. |
| **[Greedy]** \nCancer can be prevented by eating too much red meat.\n\n~~Cancer can be prevented by eating too much~~ sugar.~~\n\nCancer can be prevented by eating too much~~ fat.~~\n\nCancer can be prevented by eating too much~~ processed food.~~\n\nCancer can be...~~ |
| **[p=0.9]** More than 90% of hepatocellular carcinoma is associated with alcohol use.\n\nAbout 9 percent of cancers can be prevented if we don't eat more than 1.5 ounces of processed meat each day.\n\nDiet is believed to play a role in 7% of cancers and... |
| **[Ours]** Cancer is often treated with some combination of radiation therapy, surgery, chemotherapy and targeted therapy. Pain and symptom management are an important part of care. Palliative care is particularly important in people with advanced disease. The chance of survival depends on the type of cancer and extent of disease at the... |

Moreover, some works attempt to edit and fix the factual errors [39–41]. However, it is unclear if the improvement of fine-tuned LM for QA-style task can help to mitigate factual errors in open-ended text generation task.

**Hallucination in downstream NLG tasks** There are active efforts to reduce the unfaithfulness or factual errors of task-specific LMs fine-tuned for various downstream natural language generation (NLG) tasks such as summarization [42–48], data-to-text [49, 50, 20, 51–53] and dialogue system [54–58]. In contrast to these works, we focus on general purpose LM for open-ended text generation task.

**Human-in-the-loop** Human feedback or demonstrations are valuable to improve the factual accuracy of LMs. For example, InstructGPT [59] fine-tune the LMs with collected human feedback for a truthful generation. WebGPT [7] is trained to cite its sources when it generates output, thus allowing humans to evaluate factual accuracy by checking whether a claim is supported by a reliable source. In this work, we focus on human-free solution to mitigate nonfactual generations, as it is less expensive and easy to scale.

## 3 FACTUALITYPROMPTS and Evaluation Metrics

Our goal is to automatically measure and evaluate the factuality of large-scale pre-trained language models (LMs) for open-ended text generation. Factuality refers to being coherent to provided ground-truth knowledge sources in NLP [11]. The biggest challenge of evaluating factuality for open-ended text generation is associated with locating the ground-truth knowledge from the myriad of world knowledge. Evaluating open-ended text generation can be challenging due to the lack of ground-truth references for generation [29, 60]. In this study, the scope of our ground-truth knowledge source is set to Wikipedia [5] because this helps simplify the evaluation setup.

---

[5]Note that Wikipedia is one of the most commonly-used, accessible, large-scale, good quality, unstructured knowledge sources. Our proposed methods can easily generalize to other knowledge sources in plain text (e.g., arXiv papers, medical reports, reliable newspapers).

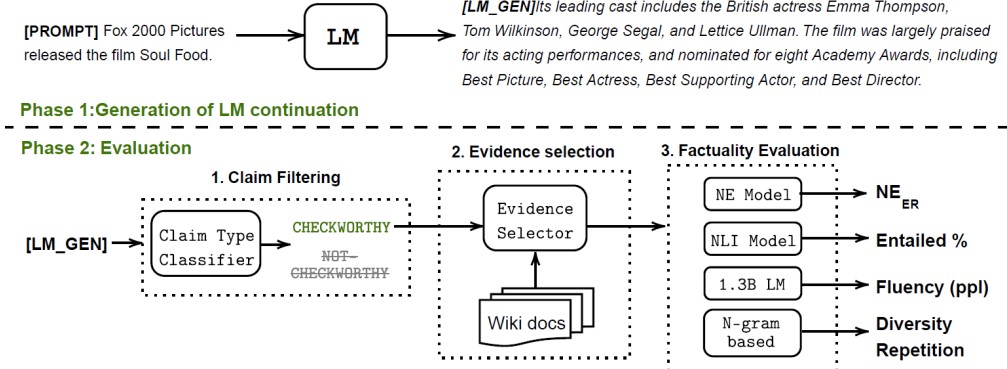

Figure 1: Illustration of our evaluation framework

As illustrated in Fig 1, our evaluation framework consists of the following phases. In phase 1, LM generates the continuations from the provided test prompts (§3.1). In phase 2, we first identify *check-worthy* continuations, which refers to the generations with facts that require factuality evaluation. One may refer to Appendix B for details. This step is necessary as open-ended text generation may generate text that does not contain facts such as personal opinion or chitchat-style text (e.g., "I like eating apples!"). Then, we prepare relevant ground-truth knowledge required for factual verification of *check-worthy* continuations (§3.2). Lastly, we calculate the factuality and quality measures (§3.3).

### 3.1 FACTUALITYPROMPTS Testset

We design our test prompts (FACTUALITYPROMPTS) that follows a similar setup as in RealToxici-tyPrompts [61], which has *toxic* and *nontoxic* prompts to evaluate the toxicity of LM continuations. FACTUALITYPROMPTS consists of *factual* and *nonfactual* prompts that allow us to study the impact of prompts' factuality on the LM continuation; this simulates the real-world scenario where input texts are not guaranteed to be factual. The data construction and statistic details are provided in Appendix D, and we will release the constructed FACTUALITYPROMPTS for future research.

### 3.2 Ground-Truth Knowledge Preparation

To evaluate the factuality of a given generation, we need to prepare relevant ground-truth knowledge. The required ground-truth knowledge can be either document-level or sentence-level, depending on the type of factuality metrics (discussed in §3.3). The correctness of factuality evaluation is crucially dependent on the correctness of the ground-truth knowledge. To ensure that our factuality evaluation is not distorted by the irrelevant provision of ground-truth knowledge, we do the following:

For **document-level** ground-truth knowledge, we directly use the Wikipedia document annotation from the FEVER dataset. This way, we can mitigate any potential error from automatic document retrieval. For **sentence-level** ground-truth knowledge, we do automatic sentence selection by using two different methods to maximize the chance of recalling the relevant ground-truth knowledge. We treat the generated text as query $q$ and Wikipedia sentences as a pool of candidates $C = \{c_1, c_2, c_3, ...c_N\}$ where $N$ is the number of sentences in the Wikipedia document. One ground-truth sentence is retrieved by obtaining TF-IDF vector representations of $q$ and $C$ and selecting the $c_i$ with the highest cosine similarity with the $q$. Another is retrieved by obtaining the contextual representation of $q$ and $C$ using SentenceTransformer [62] and selecting the $c_j$ with the highest cosine similarity.

### 3.3 Evaluation Metrics

We adapt commonly used metric designs from the hallucination literature [11]: named-entity (NE) based metric and textual entailment based metric. Each metric captures a different aspect of factuality, so we use both metrics for better understanding of factuality.

**Hallucinated NE Error**   Since NEs are one of the core building blocks of "fact", NE-related metric design is one of the common choices in literature [11, 63, 64]. In this work, we specifically adopt the NE-based metric [64] that is designed with a belief that a model is hallucinating (making factual errors) if it generates a NE that does not appear in the ground-truth knowledge source.

We define our NE-based metric to be: $NE_{ER} = |HALLU_{NE}| / |ALL_{NE}|$ where $ALL_{NE}$ is the set of all the NEs detected in the LM generation, and $HALLU_{NE}$ is subset of $NE_{All}$ that does not appear in the ground-truth Wikipedia document. Note that evaluating $NE_{ER}$ requires document-level ground-truth. To ensure the quality of the metric, we also take the same precautions used by [64]. For named entities consisting of multiple words, partial n-gram overlaps are also treated as a "match". This ensures we can address the shortened form of named entities – e.g., "Barack Hussein Obama II" vs. "Obama". Note that stopwords (e.g., the, a) are not considered in the partial n-gram overlaps. The named entities are detected using a publicly available pre-trained NE detection model from *Spacy.io*.

**Entailment Ratio** Textual Entailment (or natural language inference) is a task of determining whether a hypothesis is *entailed* by, *refuted* by, or *neutral* to a given premise [65]. Entailment-based metrics are based on the rationale that factual generation will be entailed by the ground-truth knowledge [11, 12, 66–68].

We define the entailment ratio as: $Entail_R = |ENTAIL_{gen}| / |ALL_{gen}|$, where $ALL_{gen}$ is set of all generations, and $ENTAIL_{gen}$ is the set of generations that are entailed by a entailment model. To obtain the entailment scores, we leverage a pretrained entailment model that is publicly available [6]; a RoBERTa [69] model fine-tuned on MNLI [70] dataset. $Entail_R$ requires sentence-level ground-truth because only a few Wikipedia sentences are relevant to specific factual information in a given generation. For example, "Barack Obama was born in Hawaii" is only relevant to the Wikipedia sentence that mentions his birth location. Note that our $Entail_R$ is a stricter form of metric that does not treat *neutral* class to be factual.

**Generation Quality Evaluation** We also evaluate the generation quality from three aspects: *i) Fluency* is an important aspect of text generation. We measured it by the mean perplexity of generated continuations evaluated with a large pretrained LM, which is 1.3B LM in this work . *ii) Diversity* is an important characteristic of LM that makes the generation more interesting and engaging – it is bland and boring to always generate same texts. It is measured using the mean number of distinct n-grams (we report 4-gram), normalized by the length of text [71, 72] among the 10 generations for each prompt (i.e., in total, 160,000 generations to evaluate the diversity of each method). *iii) Repetition* is a common form of degeneration that is very undesirable. We measure the number of repetitive substrings that get generated at the end of the generations by using the publicly available metric code from Holtzman et al. [29].

## 3.4 Correlation with Human Judgement

Although NE-based and entailment-based metrics have been used in downstream NLG tasks [11], they have not been utilized for evaluating factual accuracy in open-ended text generation. To ensure their validity, we collect human annotations to evaluate the correlation between our automatic factuality metrics with human judgement – i.e., are generations with higher $Entail_R$ and lower $NE_{ER}$ errors, more likely to be perceived as factual by human?

Table 2: Pearson correlation coefficients between human factuality annotation and our factuality metrics. p-values for all results are 0.00.

| Annotation | Entail$_R$ | NE$_{ER}$ |
|---|---|---|
| Expert | 0.81 | -0.77 |
| Majority-voting | 0.47 | -0.46 |

We obtained human annotations for 200 randomly chosen LM continuations of varying $NE_{ER}$ and $Entail_R$ scores. The annotators are asked to fact-check the LM continuations against Wikipedia and assign factuality label (1 = Factual : can find supporting Wikipedia evidence. 0 = Non-factual : cannot find supporting Wikipedia evidence).

The fact-checking annotation is a challenging and time-consuming task, as it requires the annotator to carefully read multiple evidences and reason over them. To improve the annotation quality, we have two types of annotations. The first type is two annotations from average English speaking workers on *Appen.com* platform, and the second type is one "expert" annotation from one of the authors who is familiar with the task and spent solid amount of time checking each samples. Based on these three annotations, we do majority voting and report the Pearson correlation results in Table 2. We also report the correlation result solely using the expert annotations, and show that there is strong correlation between human judgement of factuality and the proposed automatic metric $NE_{ER}$ and $Entail_R$. $NE_{ER}$ is negatively correlated with factuality because the lower the $NE_{ER}$ error, the better the factuality.

---

[6]Refer to the code snippet provided in https://pytorch.org/hub/pytorch_fairseq_roberta/

Table 3: The factuality of LMs with different parameter size from 12M to 530B. $NE_{ER}$ refers to the named-entity error, $Entail_R$ refers to entailment ratio, Div. refers to distinct 4-grams, and Rep. refers to repetition. ↑ means the higher the better, and ↓ means the lower the better.

| Size | Decode | Factual Prompt | | | | Nonfactual Prompt | | | |
|---|---|---|---|---|---|---|---|---|---|
| | | $NE_{ER}$↓ | $Entail_R$↑ | Div.↑ | Rep.↓ | $NE_{ER}$↓ | $Entail_R$↑ | Div.↑ | Rep.↓ |
| 126M | p=0.9 | 63.69% | 0.94% | 0.90 | 0.58% | 67.71% | 0.76% | 0.90 | 0.38% |
| | greedy | 48.55% | 8.36% | 0.03 | 59.06% | 54.24% | 6.25% | 0.03 | 59.90% |
| 357M | p=0.9 | 56.70% | 2.01% | 0.87 | 0.55% | 60.80% | 1.42% | 0.88 | 0.35% |
| | greedy | 43.04% | 14.25% | 0.03 | 45.18% | 46.79% | 9.89% | 0.04 | 46.30% |
| 1.3B | p=0.9 | 52.42% | 2.93% | 0.88 | 0.24% | 56.82% | 2.04% | 0.89 | 0.25% |
| | greedy | 39.87% | 12.91% | 0.05 | 33.13% | 45.02% | 8.75% | 0.05 | 36.20% |
| 8.3B | p=0.9 | 40.59% | 7.07% | 0.90 | 0.11% | 47.49% | 3.57% | 0.91 | 0.08% |
| | greedy | 28.06% | 22.80% | 0.07 | 19.41% | 32.29% | 15.01% | 0.07 | 13.26% |
| 530B | p=0.9 | 33.30% | 11.80% | 0.90 | 0.13% | 40.49% | 7.25% | 0.92 | 0.08% |
| | greedy | **20.85%** | **31.94%** | 0.08 | 15.88% | 27.95% | 19.91% | 0.08 | 16.28% |

## 4 Factuality Analysis of Pretrained LMs

In this section, we perform a factuality analysis of LMs from three aspects: *i)* model size, *ii)* prompt type and *iii)* decoding algorithm.

**Model Size**    Researchers have observed the trend of larger LMs outperforming smaller ones in various downstream tasks [73, 3, 2]. However, contradicting to these general observations, recent studies suggest that more misconceptions tend to be generated from larger models [31], and zero-shot fact-checking performance tend to stagnate with LM scaling [6]. We study the factuality of LMs with a range of parameter sizes (126M, 357M, 1.3B, 8.3B, 530B) to understand whether such surprising trend also applies to open-ended text generation. Note that, all LMs are pretrained on the same corpus as in [4]. As shown in Table 3, generation factuality does improve with the scaling of model size, e.g., $NE_{ER}$ drops from 63.99% to 33.30% when parameter size scales up from 126M to 530B.

**Prompt Type**    Prompts provided to the LM are known to significantly affect the quality and characteristics of LM continuations [61, 74, 75]. We use our factual and nonfactual prompts to test the behavior of LMs. Results in Table 3 show that both factual and nonfactual prompts can lead to nonfactual generations, although factual prompts always result in less nonfactual generations. Interestingly, the performance gap between factual and nonfactual prompts gets more prominent as the model size increases (4% to 7% in $NE_{ER}$ as parameter size increases from 126M to 530B). This could be due to the larger LM can better understand the prompts and imitate the factual or nonfactual prompts in the continuations.

**Decoding Algorithm**    We investigate the choice of decoding algorithms and their impacts on the factuality of generations. In particular, we compare two representative decoding algorithms that are *greedy decoding* (i.e., maximize generation likelihood) and *nucleus sampling* [29]. Nucleus sampling algorithm (a.k.a. top-$p$) samples only from the top subword candidates with total cumulative probability $p$. It is popular for open-ended text generation because it solves the degeneration problems of the greedy decoding algorithm (e.g., repetition). However, the results in Table 3 show that top-$p$ decoding underperforms greedy decoding in terms of factuality, although it obtains higher generation diversity and less repetition. This intuitively makes sense because top-$p$ can be seen as adding "randomness" to encourage diversity, which as a result, can lead to factual errors. It is important to understand that factuality of a sentence can be easily altered by one wrong choice of word. For example, "Barack Obama was born in 1961" will be nonfactual if "1961" is changed to "1962". In the same sense, greedy decoding is more factual because its way of choosing the word with the highest probability minimizes randomness and maximizes the utilization of parametric knowledge of LM [33, 36]. However, greedy decoding sacrifices generation diversity and quality.

**Error Types**    We conduct a qualitative analysis of the factual errors from greedy generation of 530B LM, to understand what are the remaining errors when the randomness from decoding choice is strictly restricted. The two notable error types were:

Table 4: **1.3B** LM results with different decoding algorithms. NE$_{ER}$ refers to named-entity error, Entail$_R$ refers to entailed class ratio, Div. refers to distinct 4-grams, and Rep. refers to repetition. ↑ means the higher, the better, and ↓ means the lower, the better. For factual-nucleus sampling, $p$, $\lambda$ and $\omega$ are nucleus probability, decay factor, and decay lowerbounds respectively. See more results with different hyperparameters in Figure 2a and 2b.

| Decoding | Factual Prompt | | | | Nonfactual Prompt | | | |
|---|---|---|---|---|---|---|---|---|
| | NE$_{ER}$↓ | Entail$_R$↑ | Div.↑ | Rep.↓ | NE$_{ER}$↓ | Entail$_R$↑ | Div.↑ | Rep.↓ |
| *Greedy* | 39.9% | 12.9% | 0.05 | 33.1% | 45.0% | 8.8% | 0.05 | 36.2% |
| *Top-p 0.9* | 52.4% | 2.9% | 0.88 | 0.2% | 56.8% | 2.0% | 0.89 | 0.3% |
| $p \mid \lambda$ | Top-$p$ + $\lambda$-decay | | | | | | | |
| 0.9 \| 0.9 | 41.1% | 10.8% | 0.43 | 30.7% | 45.7% | 6.8% | 0.47 | 34.5% |
| 0.9 \| 0.5 | 39.9% | 13.0% | 0.08 | 33.1% | 44.9% | 9.1% | 0.09 | 35.9% |
| $p \mid \lambda$ | Top-$p$ + $\lambda$-decay + $p$-reset | | | | | | | |
| 0.9 \| 0.9 | 41.5% | 10.3% | 0.52 | 10.3% | 45.4% | 6.3% | 0.57 | 9.1% |
| 0.9 \| 0.5 | 39.3% | 12.8% | 0.34 | 17.8% | 44.5% | 8.4% | 0.45 | 18.9% |
| $p \mid \lambda \mid \omega$ | Top-$p$ + $\lambda$-decay + $p$-reset + $\omega$-bound (*factual-nucleus sampling*) | | | | | | | |
| 0.9 \| 0.9 \| 0.7 | 46.2% | 5.0% | 0.78 | 1.2% | 52.2% | 3.2% | 0.80 | 0.5% |
| 0.9 \| 0.9 \| 0.3 | 42.1% | 10.1% | 0.55 | 7.1% | 46.5% | 5.6% | 0.59 | 6.4% |
| 0.9 \| 0.9 \| 0.2 | 41.7% | 9.9% | 0.52 | 8.6% | 45.6% | 6.2% | 0.56 | 7.6% |
| 0.9 \| 0.5 \| 0.3 | 41.0% | 12.2% | 0.47 | 13.0% | 46.0% | 7.0% | 0.51 | 12.7% |
| 0.9 \| 0.5 \| 0.2 | 39.3% | 12.8% | 0.38 | 16.1% | 45.2% | 7.8% | 0.42 | 16.9% |

- **Named Entity Mix-up**: Mixing up similar types of the named entity. For example, LM generated "*The movie is based on the novel of the same name by Gayle Forman.*" about a film called "*The Best of Me*". However, the correct author's name is "Nicholas Sparks", not "Gayle Forman". Note that Gayle Forman is also an American young adult fiction author who writes similar type of novels as Nicholas Sparks.
- **Fabricated Fact:** Fabricating some random facts. For example, "*Samuel Witwer's father is a Lutheran minister.*" Note that, the pretraining corpus contains non-factual or fictional information, which can also contribute to such fabricated facts.

Both error types can be viewed as wrong associations of entities that appear at different parts of the training corpus with similar context. Such behavior is unsurprising because these LMs are uniformly trained with the next subword prediction objective instead of a fact-related objective.

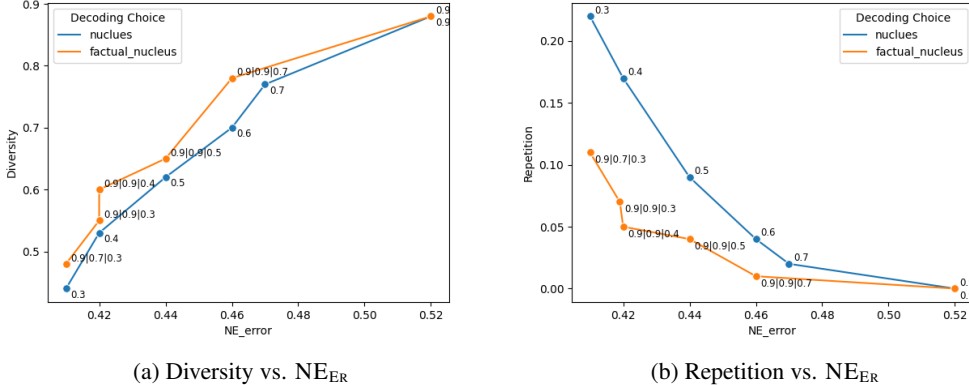

(a) Diversity vs. NE$_{ER}$     (b) Repetition vs. NE$_{ER}$

Figure 2: Comparison between nucleus sampling (blue line) and factual-nucleus sampling (orange line). The x-axis is named entity error NE$_{ER}$. The y-axes are diversity and repetition in (a) and (b) respectively. The lower the repetition, the better. It is evident that factual-nucleus sampling has better trade-offs between factuality and diversity/repetition. For a reference, the diversity score of randomly sampled 5000 Wikipedia documents is 0.767.

# 5 Factual-Nucleus Sampling

In this section, we propose a new sampling algorithm that achieves a better trade-off between generation quality and factuality than existing decoding algorithms.

## 5.1 Method

We hypothesize that the randomness of sampling is more harmful to factuality when it is used to generate the latter part of a sentence than the beginning of a sentence. There is no preceding text at the start of a sentence, so it is safe for LM to generate anything as long as it is grammatical and contextual. However, as the generation proceeds, the premise become more determined, and fewer word choices can make the sentence factual. Given the example "*Samuel Witwer's father is a Lutheran minister*", the beginning of the sentence "*Samuel Witwer's father is*" is not nonfactual. However, the continuation of "*Lutheran minister*" makes the sentence nonfactual. Therefore, we introduce the *factual-nucleus sampling* algorithm that dynamically adapts the "nucleus" $p$ along the generation of each sentence. In *factual-nucleus sampling*, the nucleus probability $p_t$ to generate the $t$-th token within each sentence is,

$$p_t = \max\{\omega, \ p \times \lambda^{t-1}\},$$

where $\lambda$ is the decay factor for top-$p$ probability, and $\omega$ lower bounds the decay of probability. Specifically, it has the following parts:

- $\lambda$-**decay**: Given that top-$p$ sampling pool is selected as a set of subwords whose cumulative probability exceeds $p$, we gradually decay the $p$ value with decay factor $\lambda$ at each generation step to reduce the "randomness" through time.

- $p$-**reset**: The nucleus probability $p$ can quickly decay to a small value after a long generation. So, we reset the $p$-value to the default value at the beginning of every new sentence in the generation (we identify the beginning of a new sentence by checking if the previous step has generated a full-stop). This reduces the unnecessary cost of diversity for any long generations.

- $\omega$-**bound**: If $\lambda$-decay is applied alone, the $p$-value could become too small to be equivalent to greedy decoding and hurt diversity. To overcome this, we introduce a lower-bound $\omega$ to limit how far $p$-value can be decayed.

We will show the importance of each parts with ablation studies.

## 5.2 Result

We report our decoding experimental results with 1.3B LM [7] in Table 4. Additions of $\lambda$-decay helps improve top-$p$ 0.9 factuality results – for instance, with decay rate $\lambda = 0.5$, there is 12.5% drop in $\text{NE}_{\text{ER}}$ and 10.1% gain in $\text{Entail}_{\text{R}}$. However, this affects the diversity and repetition to become similar to greedy decoding. $p$-reset mitigates the repetition issue and improves diversity metric without losing much in factuality metric. The effect is more drastic for the $\lambda = 0.5$ option, where it achieves 0.26 gains in diversity metric with negligible changes in factuality scores. By also adding $\omega$-bound, we obtain the anticipated factuality performance (i.e., similar to greedy decoding), with great improvement in generation quality over greedy; with $p$=0.9, $\lambda$=0.9, $\omega$=0.3, we achieve $\times 11$ improvement in diversity and $\times 4.6$ improvement in repetition over greedy. Although our factual-nucleus sampling still under-performs top-$p$ 0.9 in terms of diversity, we believe this is an acceptable trade-off to improve the factuality of LM for factually sensitive open-ended generation tasks. Our proposed decoding does not harm the sentence fluency; its perplexity do not exceed the perplexity of top-p. Refer to Appendix F for full perplexity results.

To further illustrate the underlying trade-off, we also compare the proposed factual-nucleus sampling against the nucleus sampling with lower $p$ values that are also expected to have lower randomness, thus less factual error, in generations. Specifically, we plotted results for nucleus sampling with $p = \{0.9, 0.7, 0.6, 0.5, 0.4, 0.3\}$, and factual nucleus sampling with the following $p \mid \lambda \mid \omega$ choices: 0.9|0.9|0.7, 0.9|0.9|0.5, 0.9|0.9|0.4, 0.9|0.9|0.3, 0.9|0.7|0.3. The Fig 2a and Fig 2b respectively show that the factual nucleus sampling method has better trade-offs than top-$p$ in factuality-vs-diversity and factuality-vs-repetition. In other words, it always achieves better factuality score with the same level of diversity and repetition scores.

---

[7] 1.3B LM is mainly used as it is big enough to have good learning capacity yet not too resource expensive.

# 6 Factuality-Enhanced Continued Training

This section introduces factuality-enhanced method for continued training of LMs [30]. We introduce the TOPICPREFIX for better awareness of facts and the sentence completion loss as training objective.

## 6.1 Prepending TOPICPREFIX

Unstructured factual knowledge typically exists at a document level (i.e., a group of factual sentences about an entity). This means that sentences can contain pronouns (e.g., she, he, it), making these sentences factually useless standalone. To illustrate with an example from Barack Obama's Wikipedia page, "He previously served as a U.S. senator from Illinois from 2005 to 2008" cannot be a useful standalone fact because it is unclear who "He" is. Due to the GPU memory limit and computation efficiency, it is common to chunk documents in LM training corpus. This causes the "fragmentation" of information and leads to wrong associations of entities that appear in independent documents with similar contexts. As a remedy, we propose to prepend TOPICPREFIX to sentences in the factual documents to make each sentence serve as a standalone fact. In our experiments, we mainly utilize Wikipedia as the factual corpus and the Wikipedia document name as the TOPICPREFIX.

## 6.2 Sentence Completion Loss

We propose a sentence completion loss to address the incorrect association learned between entities. To explain our rationale, let us recall the nonfactual example from §5: "*Samuel Witwer's father is a Lutheran minister*". This sentence is nonfactual because LM failed to generate factually correct information after "*is*". In other words, LM failed to accurately *complete* the sentence given the generated context. One reason is that the LM is uniformly trained to predict each subword token within the sentence, when ensuring the correct prediction at the latter section of sentence is more critical for factuality. Therefore, we construct a sentence completion loss, which makes the LM focus on predicting the subwords later in the sentence. For implementation, we determine a pivot $t$ for each sentence, and then apply zero-masking for all token prediction losses before $t$. This pivot is only required during training (i.e., no pivot needed during inference time).

We emphasize that this loss masking is different from the input token masking applied in BERT [73] or BART [76], and the LM is still trained in an autoregressive manner. Note that many BART-based summarization models are known to still suffer from factual errors, suggesting that masked prediction at the encoder level may not effectively transfer well to autoregressive text generation.

In this work, we explore three strategies (from simple to complex) to determine the pivot $t$:

- $SC_{\text{HALF}}$: pivot $t = 0.5 \times$ sentence-length.
- $SC_{\text{RANDOM}}$: random pivot, e.g., $t \sim \text{uniform}[0.25, 0.75] \times$ sentence-length.
- $SC_{\text{ROOT}}$: pivot $t =$ position of ROOT (relation) from dependency parsing.

Our experiments show that the simplest $SC_{\text{HALF}}$ performs on par with the complex ones (such as $SC_{\text{ROOT}}$), thus, we suggest future work to choose $SC_{\text{HALF}}$ strategy.

## 6.3 Results

The results are reported in Table 5, and experimental setups are reported in Appendix C.

**Inefficiency of Domain Adaptive Training** The pre-training corpus of LM contains both factual texts (e.g., Wikipedia) and potentially nonfactual texts (e.g., rumors, fake news) [8]. The nonfactual domain of the training corpus could be the problem. Thus, we conduct a baseline experiment that does domain-adaptive training with strictly factual domain text only (i.e., Wikipedia). Interestingly, we find that domain-adaptive training can hardly improve generation factuality.

**Effect of TOPICPREFIX** Continued pre-training of 1.3B LM with TOPICPREFIX preprocessed Wikipedia alone can already improve the factuality, especially in terms of $NE_{\text{ER}}$. For example, it reduces the $NE_{\text{ER}}$ from $42.1\%$ to $27.6\%$ when we use the factual-nucleus decoding $(0.9 \mid 0.9 \mid 0.3)$, which even outperforms the 1.3B with greedy decoding ($NE_{\text{ER}}$: $27.6\%$ vs. $39.9\%$) with much less repetition ($8.0\%$ vs. $33.1\%$).

**Effect of Sentence Completion Loss** The proposed sentence completion loss further helps to improve the factuality, especially for the $\text{Entail}_{\text{R}}$. For example, when one uses factual-nucleus

---

[8]See [4] for details of pre-training corpus.

Table 5: Results for factuality enhanced training. The decoding settings are formatted as: nucleus probability $p$, decay rate $\lambda$, lower-bound $\omega$.

| Decoding | Factual Prompt | | | | Nonfactual Prompt | | | |
|---|---|---|---|---|---|---|---|---|
| $(p \mid \lambda \mid \omega)$ | $NE_{ER}\downarrow$ | $Entail_R\uparrow$ | Div. | Rep. | $NE_{ER}$ | $Entail_R$ | Div. | Rep. |
| Vanilla Pretrained LM (1.3B) | | | | | | | | |
| 0.9 | 52.4% | 2.9% | 0.88 | 0.2% | 56.8% | 2.0% | 0.89 | 0.3% |
| 0.9 \| 0.9 \| 0.3 | 42.1% | 10.1% | 0.55 | 7.1% | 46.5% | 5.6% | 0.59 | 6.4% |
| Factual Domain-Adaptive Training with Wikipedia (1.3B) | | | | | | | | |
| 0.9 | 52.5% | 2.8% | 0.85 | 0.2% | 55.8% | 2.2% | 0.86 | 0.1% |
| 0.9 \| 0.9 \| 0.3 | 42.7% | 7.1% | 0.51 | 7.2% | 48.2% | 4.9% | 0.56 | 6.0% |
| TOPICPREFIX (1.3B) | | | | | | | | |
| 0.9 | 34.4% | 4.2% | 0.84 | 0.3% | 36.2% | 2.7% | 0.85 | 0.2% |
| 0.9 \| 0.9 \| 0.3 | 27.6% | 8.7% | 0.43 | 8.0% | 30.5% | 6.1% | 0.47 | 6.9% |
| TOPICPREFIX + $SC_{ROOT}$ (1.3B) | | | | | | | | |
| 0.9 | 32.5% | 6.7% | 0.83 | 1.2% | 34.3% | 4.6% | 0.84 | 1.1% |
| 0.9 \| 0.9 \| 0.3 | 24.7% | 15.8% | 0.40 | 13.6% | 27.6% | 9.1% | 0.44 | 13.7% |
| TOPICPREFIX + $SC_{RANDOM}$ (1.3B) | | | | | | | | |
| 0.9 | 32.0% | 7.9% | 0.81 | 1.2% | 34.2% | 5.5% | 0.83 | 1.1% |
| 0.9 \| 0.9 \| 0.3 | 23.6% | 17.6% | 0.39 | 14.2% | 26.9% | 9.3% | 0.42 | 13.2% |
| TOPICPREFIX + $SC_{HALF}$ (1.3B) | | | | | | | | |
| 0.9 | 31.6% | 7.6% | 0.81 | 1.4% | 33.5% | 5.1% | 0.83 | 1.5% |
| 0.9 \| 0.9 \| 0.3 | 23.6% | 17.4% | 0.38 | 14.4% | 27.2% | 10.2% | 0.42 | 13.1% |
| Vanilla Pretrained LM (530B) | | | | | | | | |
| 0.9 | 33.3% | 11.8% | 0.90 | 0.1% | 40.5% | 7.25% | 0.92 | 0.1% |
| TOPICPREFIX + $SC_{HALF}$ (530B) | | | | | | | | |
| 0.9 | 18.3% | 19.3% | 0.68 | 0.1% | 21.7% | 13.7% | 0.68 | 0.1% |
| 0.9 \| 0.9 \| 0.3 | **14.5%** | **25.5%** | 0.33 | 0.2% | **17.7%** | **20.0%** | 0.33 | 0.1% |

decoding on trained 1.3B model, TOPICPREFIX + $SC_{HALF}$ can further improve $Entail_R$ from $8.7\%$ to $17.4\%$ than TOPICPREFIX alone, while reducing $NE_{ER}$ from $27.6\%$ to $23.6\%$. Note that the results show consistent improvement across different pivot selection strategies, suggesting that the sentence completion loss is robust. In particular, the simplest $SC_{HALF}$ performs as good as others or even better in terms of several metrics. Thus we recommend it as the default option.

**530B vs 1.3B** As expected, our method on 530B LM further reduces the factual errors and achieves the lowest $NE_{ER}$ ($14.5\%$) and the highest $Entail_R$ ($25.5\%$). Surprisingly, our method on 530B LM lead to less diverse generation than 1.3B LM despite the significant improvement in the generation quality (i.e., near perfect repetition scores $0.1\%$ $0.2\%$). We conjecture that this is the trade-off between the factuality and diversity for 530B LM.

## 7 Conclusion

In this work, we establish a benchmark to measure and analyze factuality in open-ended text generation tasks. We propose *factual-nucleus sampling* that improves generation factuality at inference time, and the combination of sentence completion loss and TOPICPREFIX pre-processing that improves factuality with continued training. We demonstrate that our methods are effective in improving the factuality. Lastly, our results shed light on the existence of the trade-off between diversity and factuality. We strongly believe this is an important insight that will help researchers make a better-informed decision about their model design - i.e., appropriately prioritize the desirable attribute of their LM (factuality vs. diversity) according to the final goal of their task. Potential future work would be to reduce the degree of the observed trade-offs.

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
