# A  Generation Examples

We provide more generation examples from the pretrained 530B LM with greedy and top-$p$ sampling ($p = 0.9$), and factuality-enhanced 530B LM with factual-nucleus sampling (Ours). Green indicates factual, red indicates nonfactual, and ~~striked text~~ indicates repetition. Refer to Appendix G for more examples. Disclaimer: Authors tried to exhaustively check the factuality of the following generations, however, there is no 100% guarantee about the annotations.

---

**[Category]** Medical
**[Factual Prompt]** One of lung cancer's symptoms is shortness of breath.

---

**[Greedy]** The American Cancer Society estimates that more than 228,000 people will be diagnosed with lung cancer in the United States this year.\n\n ~~The American Cancer Society estimates that more than 228,000 people will be diagnosed with lung cancer in the United States this year.\n\nThe American Cancer Society estimates that more than 228,000 people will be diagnosed with lung cancer in the United...~~

**[p=0.9]** That would make an oxygen mask one of the more popular treatments for this devastating disease. It helps ease breathing and give patients back their strength. Learn more here.<|endoftext|>"

**[Ours]** Other symptoms include coughing (including coughing up blood), chest pain, hoarseness, weight loss, and loss of appetite. The symptoms of lung cancer may resemble those of a bad cold or pneumonia. A chest x - ray is necessary to confirm the diagnosis. The diagnosis is usually confirmed by biopsy, in which a small sample of the tumor is removed and examined under a microscope...

---

**[Category]** Actor
**[Factual Prompt]** Leonardo DiCaprio played a supporting role in What's Eating Gilbert Grape.

---

**[Greedy]** The film was shot in the town of Endora, Iowa.\n\n~~The film was shot in the town of Endora, Iowa.\n\nThe film was shot in the town of Endora, Iowa.\n\nThe film was shot in the town of Endora, Iowa.\n\nThe film was shot in the town of Endora, Iowa.\n\nThe film was shot in the town of Endora, Iowa.\n\nThe film was shot in the town of Endora, ...~~

**[p=0.9]** DiCaprio was desperate to have the lead role in the film. However, after realizing the amount of time that was required to learn all the lines, he gave the role to Johnny Depp instead.\n\nTitanic (1997)\n\nLeonardo DiCaprio and Kate Winslet were paid the same amount for their roles in Titanic.\n\nAlthough DiCaprio was relatively unknown when the film was released, Paramount Pictures paid him and Winslet the same amount.

**[Ours]** In 1993 , he played the mentally handicapped brother of Johnny Depp 's character in " What's Eating Gilbert Grape" , a comic - tragic odyssey of a dysfunctional Iowa family. Director Lasse Hallström admitted he was initially looking for a less good - looking actor but finally settled on DiCaprio as he had emerged as "the most observant actor" among all who auditioned . Budgeted at US$ 11 million , the film became a critical success, resulting in various accolades for DiCaprio , who was awarded the National Board of Review Award and nominated for both an Academy Award and a Golden Globe Award for his portrayal of Arnie Grape...

---

**[Category]** Company
**[Factual Prompt]** KFC is the second biggest restaurant chain in the world.

---

**[Greedy]** The company is owned by Yum! Brands, which also owns Pizza Hut and Taco Bell.\n\nKFC is known for its fried chicken, but it also offers other food items like chicken sandwiches, wraps, salads, and even rice bowls.\n\n~~KFC is known for its fried chicken, but it also offers other food items like chicken sandwiches, wraps, salads, and even rice bowls.\n\nKFC is known for its fried chicken, but it also offers other food items like chicken sandwiches, wraps, salads, and even rice bowls.\n\nKFC...~~

**[p=0.9]** While many restaurants worldwide have been growing their mobile commerce revenues, KFC chose to implement its payment system via cash. Customers must visit a traditional cash register to buy food.\n\nThe order-and-pay system that KFC had been using at its stores didn't allow customers to pay with their smartphones. Instead, customers have to stand in long lines to order and pay for their meals. The process involves taking a physical receipt, which is then placed at the counter, to process the order.\n\n...

**[Ours]** KFC was founded by Colonel Harland Sanders , an entrepreneur who began selling fried chicken from his roadside restaurant in Corbin , Kentucky , during the Great Depression . Sanders identified the potential of the restaurant franchising concept , and the first " Kentucky Fried Chicken " franchise opened in Utah in 1952 . KFC popularized chicken in the fast food industry , diversifying the market by challenging the established dominance of the hamburger . By branding himself as " Colonel Sanders " , Harland became a prominent figure of American cultural history , and his image remains widely used in KFC advertising to this day...

---

**[Category]** History
**[Factual Prompt]** San Francisco was first colonized by Spanish colonists.

**[Greedy]** The Spanish arrived in 1776 and established the Presidio of San Francisco in 1776. The Mission San Francisco de Asís was founded in 1776. The Mission was the first European settlement in the city. ~~The Mission was the first European settlement in the city. The Mission was the first European settlement in the city. The Mission was the first European settlement in the city. The Mission was the first European settlement in the city. The Mission was the first European settlement in the city.~~

**[p=0.9]** A Spaniard, José Joaquin Moraga, who accompanied the De Anza expedition, named the bay, which he called "Puerto de San Francisco", on or about October 30, 1776. On January 12, 1777, the Presidio of San Francisco, founded by Moraga, was established by the Spanish. The Mission San Francisco de AsŎ0eds, established in the same year, was the first Mission on the Pacific Coast of North America. In 1781, the settlers, numbering about 200...

**[Ours]** The Presidio of San Francisco , a Spanish military outpost , was established in 1776 . It was the first European settlement in what is now the city . The Mission San Francisco de Asís ( Mission Dolores ) , established in 1776 , was the city 's first major settlement for the next half - century . The California Gold Rush of 1849 brought rapid growth , making it the largest city on the West Coast at the time . San Francisco became a consolidated city - county in 1856 . San Francisco 's status as the West Coast 's largest city peaked between 1870 and 1900 , when around 25 % of California 's population resided in the city proper . After three - quarters of the city was destroyed by the 1906 earthquake and fire , San Francisco was quickly rebuilt...

**[Category]** Sports
**[Factual Prompt]** Michael Jordan helped popularize the NBA in the 80s and 90s.

**[Greedy]** The NBA is the most popular basketball league in the world.\n\n~~The NBA is the most popular basketball league in the world.\n\nThe NBA is the most popular basketball league in the world.\n\nThe NBA is the most popular basketball league in the world.\n\nThe NBA is the most popular basketball league in the world.\n\nThe NBA is the most popular basketball league in the world.\n\nThe NBA is the most...~~

**[p=0.9]** He was named the most valuable player in the NBA Finals six times. But the former Chicago Bulls star and current Charlotte Hornets owner has another MVP title to add to his list: MVP of Valentine's Day. An online sports book claims that Jordan is the most searched player of all time on Feb. 14.\n\n » Read more trending news\n\nFanDuel Sportsbook, an online gaming company, said it reviewed searches on its site for the last seven years and discovered that Jordan is the most searched-for NBA player on Valentine's Day, Forbes",

**[Ours]** Jordan 's individual accolades and accomplishments include six NBA Finals Most Valuable Player ( MVP ) Awards , ten scoring titles ( both all - time records ) , five MVP Awards , ten All - NBA First Team designations , nine All - Defensive First Team honors , fourteen NBA All - Star Game selections , three All - Star Game MVP Awards , three steals titles , and the 1988 NBA Defensive Player of the Year Award. He holds the NBA records for highest career regular season scoring average ( 30.12 points per game ) and highest career playoff scoring average ( 33.45 points per game ). Jordan is also known for his product endorsements . He fueled the success of Nike 's Air Jordan sneakers , which were introduced in 1984...

**[Category]** TV
**[Factual Prompt]** In Too Deep is an American crime thriller film.

**[Greedy]** It was directed by Michael Rymer and released in 1999. The film stars Omar Epps, LL Cool J, Nia Long, Stanley Tucci, and Giancarlo Esposito. A young man named Michael (Omar Epps) is released from prison after serving a sentence for drug dealing. He is determined to go straight, but his friend Tommy (LL Cool J) is not so sure. Tommy is a drug dealer who has been working for a crime boss named Dwayne Gittens (Stanley Tucci). Gittens is a violent man who will not hesitate to kill anyone who crosses him.\n\n Michael is soon dragged back into the drug world when Tommy is arrested for killing a man who tried to rob him. Gittens tells Michael that he will free Tommy if Michael will sell a large amount of drugs...

**[0.9]** The plot concerns the federal government's attempts to lure a bank robber out of hiding by teaming up a convicted high-profile criminal with a U.S. Marshal. It stars Omar Epps and LL Cool J and is directed by Michael Rymer.\n\nThe film is also known as Deeper and In Too Deep. The film had a DVD release on December 26, 2000, and an HD DVD release on April 30, 2007.\n\nPlot\nRonnie Pullman (Omar Epps) is a thief who has made a name for himself by stealing from...

**[Ours]** It was directed by Michael Rymer from a screenplay written by Michael Henry Brown and Paul Aaron . The film stars Omar Epps , LL Cool J , Stanley Tucci , Pam Grier and Nia Long. Jeffrey Cole (Omar Epps) is a recent graduate of the Cincinnati police academy who dreams of working undercover. Cole manages to get an undercover assignment the day of graduation and earns the praise of his superiors . He is soon given the task of taking down statewide crack dealer Dwayne Gittens ( LL Cool J ) , an underworld boss so powerful that his nickname is "God"

| | |
|---|---|
| **[Category]** Medical | |
| **[Non-Factual Prompt]** Obesity and endometrial cancer have nothing to do with each other. | |

**[Greedy]** \n\nObesity is a risk factor for endometrial cancer. \n\nObesity is not a risk factor for endometrial cancer.\n\nCorrect\n\nIncorrect\n\nQuestion 3 of 10\n\n3. Question\n\nWhich of the following statements is true?\n\nEndometrial cancer is the most common gynecologic cancer in the United States.\n\nEndometrial cancer is the most common gynecologic cancer in the United States.\n\nEndometrial cancer is the most common gynecologic cancer in the United States.\n\nEndometrial cancer is the most common gynecologic cancer in the United States.

**[p=0.9]** \n\nThen why does it get such a high incidence rate with obese women?\n\nIt doesn't. The problem is that fat women are more likely to have abnormal uterine bleeding.\n\nAnd with an enlarged uterus, it may be difficult for a physician to detect a...

**[Ours]** Obesity increases the risk of endometrial cancer . This is thought to be due to higher levels of estrogens produced by the adipose tissue . Obesity also causes changes in the endometrium that make it more susceptible to cancer...

---

**[Category]** Sports
**[Non-Factual Prompt]** Michael Jordan was inactive in the 1980s and 90s.

**[Greedy]** The Chicago Bulls were inactive in the 1980s and 90s.\n\nThe Chicago Bulls were inactive in the 1980s and 90s.\n\nThe Chicago Bulls were inactive in the 1980s and 90s.\n\nThe Chicago Bulls were inactive in the 1980s and 90s...

**[p=0.9]** This caused concern that the new coach was not doing anything right. But there was still hope because he promised the following: We're going to be active traders on the phones. Except it turned out he really meant they were going to be doing a lot of long distance calling and give...

**[Ours]** Jordan 's first retirement in 1993 was very short - lived , as he returned to the NBA in March 1995 , wearing the number 45 , after his father was murdered .\n The Bulls were then eliminated in the second round of the playoffs by the Orlando Magic.\n Jordan was suspended by the NBA for...

---

**[Category]** TV
**[Non-Factual Prompt]** Modern Family is solely a Dutch television sitcom.

**[Greedy]** It is the first Dutch television series to be produced in HD. The series is produced by the Dutch broadcasting company Talpa, which had previously produced the successful series Tienstok. The series is written by Dutch screenwriter and director Joram Lürsen.\n\nThe series is about a...

**[p=0.9]** It follows the story of a quite stereotypical Dutch family (although unlike the Dutch, the family in this sitcom has only two kids and only the youngest is still a minor, whereas in reality most Dutch families have at least three kids). The majority of the episodes are just ordinary family life, but the writers do not...

**[Ours]** The show premiered on Dutch television on 18 September 2007 on RTL 4 . The show was originally created by Steve Levitan and Christopher Lloyd , who also serve as executive producers . The series is set in the fictional suburban town of Dunphy, California . The show follows the lives of the Dunphy family , consisting of Phil Dunphy ( Eric Stonestreet ) , Claire Dunphy ( Julie Bowen ) , Cameron Dunphy ( Jesse Tyler Ferguson ) , Mitchell Dunphy ( Ty Burrell ) , and Lily Dunphy ( Sofia Vergara )...

---

**[Category]** Singer
**[Non-Factual Prompt]** Michael Jackson made his debut with his sister Janet.

**[Greedy]** \n\nThe Jackson 5 made their debut on the show in 1969.\n\nThe Jackson 5 made their debut on the show in 1969.\n\nThe Jackson 5 made their debut on the show in 1969.\n\nThe Jackson 5 made their debut on the show in 1969.\n\nThe Jackson 5...

**[p=0.9]** Two performances followed before Jackson would begin a solo career, while still continuing to perform with his brothers in The Jacksons.\n\n1977\n\nMichael Jackson releases his first solo album 'Off the Wall'. A critical and commercial success, 'Off the Wall' reached No.5 in the UK album...

**[Ours]** The Jackson 5 won several awards within the black music community , including the American Music Award for Favorite Soul / R&B Band / Duo in 1970 and 1972 . Jackson 's solo career began in 1971 while at Motown with his first solo album , " Got to Be There " , which was certified platinum . His second album , " Ben " , was released in 1972 . The album sold over a million copies , and the single " Ben " reached number one on the " Billboard " Hot 100 . Jackson 's third album , " Music & Me " , was released in 1973 . In 1975 , Jackson moved to Epic Records...

# B  Details about Claim Filtering Step in §3

The goal of open-ended text generations does not require all generations to always contain "facts". There can be generations that are perfectly grammatical and fluent, yet do not contain any checkworthy content such as personal opinions and daily small talks. Thus, we filter out "not-checkworthy" sentences that possess any of the following characteristics:

- Contains no named entities, which are important building blocks of fact or information. E.g., "Check this out", "To say that a person is an example of something is absurd."
- Contains first-person pronouns (i.e., I, we, and us), which are strong signal for personal opinions or casual chitchat style of writing. E.g., "I think...", "I believe..."
- Contains question mark. E.g., "Do you want to hear something interesting?", "Did you know?", "What are your thoughts?"

# C  Experiment Details

**Usage Example of TOPICPREFIX**   Here, we provide an example of how the training corpus looks like when TOPICPREFIX is applied.

The following Wikipedia paragraph about Barack Obama:

*Barack Hussein Obama II (born August 4, 1961) is an American politician who served as the 44th president of the United States from 2009 to 2017. He was the first African-American president of the United States. A member of the Democratic Party, he previously served as a U.S. senator from Illinois from 2005 to 2008 and as an Illinois state senator from 1997 to 2004.*

is transferred into:

*Barack Obama ==> Barack Hussein Obama II (born August 4, 1961) is an American politician who served as the 44th president of the United States from 2009 to 2017. Barack Obama ==> He was the first African-American president of the United States. Barack Obama ==> A member of the Democratic Party, he previously served as a U.S. senator from Illinois from 2005 to 2008 and as an Illinois state senator from 1997 to 2004. .*

**Hyper-parameters and training details**   The hyper-parameters for 1.3B factuality enhancement training were: learning rate 2e-6, batch size 64, maximum input sequence length 2048. For 530B model, the hyper-parameters were: learning rate 1e-5, batch size 512, maximum input sequence length 2048. The architecture details of pre-trained LMs are in Table 6. During inference, we set maximum subword sequence length to be 150. Same Wikipedia corpus with topic-prefix is commonly used for all our factuality enhancing training.

**Detail about pre-trained LMs**   All LMs with different sizes are pre-trained on the same corpus, following the experimental details in [76].

Table 6: Architecture details of pre-trained LMs.

| Models (#/parameters) | #/layers | #/hidden size | #/ attention heads |
|---|---|---|---|
| 126M | 12 | 768 | 12 |
| 357M | 24 | 1024 | 16 |
| 1.3B | 24 | 2048 | 32 |
| 8.3B | 40 | 4096 | 64 |
| 530B | 105 | 20480 | 128 |

# D  FACTUALITYPROMPTS Details

Since high quality fact-related data collection requires a lot of human efforts, we instead utilize a well-established fact-related dataset, FEVER [77], to construct our factual and nonfactual prompts. FEVER is a fact-checking dataset consisting of claims that are SUPPORTED, REFUTED or unverifiable (NOTENOUGHINFO) by Wikipedia documents. These claims are created by annotators who were asked to alter or paraphrase the sentences from Wikipedia. We leverage the SUPPORTED and REFUTED

Table 7: Data statistics of FACTUALITYPROMPTS

|  | Factual Prompts | Nonfactual Prompts |
|---|---|---|
| # Prompts | 8000 | 8000 |
| Avg # Tokens | 9.77 | 9.48 |

claims from FEVER validation set [9] as the factual and nonfactual prompts, respectively. To further ensure the quality of the test set, we filter out claims that are not appropriate to serve as prompts – e.g., extremely short claims that are not enough to provide any context to the LM. The data statistics after filtering is reported in Table 7.

# E    Limitations and Societal Impact

Although the factual-nucleus sampling requires the same amount of computation as regular top-$p$ sampling, the continued pre-training of large language models will have some negative carbon footprint. However, our task itself (trying to improve factuality) will bring more overall benefit to the community and society, by allowing the language models to generate less fake information and be safer for deployment. In terms of ethical consideration, to the best of our knowledge, Wikipedia has no private personal information or any inappropriate content (problematic discrimination towards particular demographic groups, NSFW contents, hate speech, etc). So, fine-tuning our model on it will not encourage unfairness, biases or toxic output.

# F    Extended Experimental Results

## F.1    Ablation Study of Sentence Completion Loss

A small scale experiments using 3000 Factual Prompts are conducted to explore the stand-alone impact of sentence completion loss. As shown in Table 8, the only having sentence completion loss is indifferent to having the standard factual-domain adaptive training (i.e., negligible difference in factuality). However, when used together with TOPICPREFIX, it results in a significant boost for both factuality metrics.

Table 8: Ablation Study of Sentence Completion Loss

| Model Choice | Decoding | $NE_{ER}\downarrow$ | $Entail_R\uparrow$ |
|---|---|---|---|
| Default Wiki FT baseline | 0.9 | 55.92% | 2.58% |
|  | 0.9 \| 0.9 | 45.78% | 7.12% |
| $SC_{ROOT}$ | 0.9 | 55.81% | 2.36% |
|  | 0.9 \| 0.9 | 44.80% | 6.39% |
| $SC_{ROOT}$ + TopicPrefix | 0.9 | 35.15% | 6.67% |
|  | 0.9 \| 0.9 | 26.04% | 15.67% |
| $SC_{HALF}$ | 0.9 | 56.16% | 2.08% |
|  | 0.9 \| 0.9 | 45.13% | 6.64% |
| $SC_{HALF}$ + TopicPrefix | 0.9 | 34.18% | 7.63% |
|  | 0.9 \| 0.9 | 25.15% | 18.58% |
| $SC_{NE}$ | 0.9 | 56.42% | 1.88% |
|  | 0.9 \| 0.9 | 45.84% | 6.97% |
| $SC_{NE}$ + TopicPrefix | 0.9 | 35.87% | 4.59% |
|  | 0.9 \| 0.9 | 28.87% | 10.71% |

---

[9]The testset is not publicly released and can only be accessed through the FEVER workshop submission site. Therefore, it is common practice to leverage validation set instead.

## F.2 Experimental Results with Perplexity

In this subsection, we provide experimental results including the perplexity scores (PPL) of generated text evaluated on the 1.3B pretrained LM as a *fluency* measure. The results consistently indicate that our proposed decoding and training methods do not harm the fluency of the generation. For instance, in Table 9, all our decoding choices result in PPL scores between $1.9 \sim 4.1$ that are smaller than Top-$p$ 0.9 PPL score 12.0.

To provide full details about the columns reported in Table 9 and Table 10, $NE_{ER}$ refers to the named-entity error, $Entail_R$ refers to entailment ratio, Div. refers to distinct 4-grams and Rep. refers to repetition. $\uparrow$ means the higher the better, and $\downarrow$ means the lower the better.

Table 9: The factuality of **1.3B** LM with different decoding algorithms. $p$ is the nucleus probability, $\lambda$ is the decay factor, and $\omega$ lower bounds the decay.

| Decoding | Factual Prompt | | | | | Nonfactual Prompt | | | | |
|---|---|---|---|---|---|---|---|---|---|---|
| | $NE_{ER}\downarrow$ | $Entail_R\uparrow$ | Div.$\uparrow$ | Rep.$\downarrow$ | PPL$\uparrow$ | $NE_{ER}\downarrow$ | $Entail_R\uparrow$ | Div.$\uparrow$ | Rep.$\downarrow$ | PPL$\uparrow$ |
| *Greedy* | 39.9% | 12.9% | 0.05 | 33.1% | 1.9 | 45.0% | 8.8% | 0.05 | 36.2% | 2.0 |
| *Top-p 0.9* | 52.4% | 2.9% | 0.88 | 0.2% | 10.9 | 56.8% | 2.0% | 0.89 | 0.3% | 12.0 |
| $p \mid \lambda$ | Top-$p$ + $\lambda$-decay | | | | | | | | | |
| 0.9 \| 0.9 | 41.1% | 10.8% | 0.43 | 30.7% | 2.02 | 45.7% | 6.8% | 0.47 | 34.5% | 2.13 |
| 0.9 \| 0.5 | 39.9% | 13.0% | 0.08 | 33.1% | 1.89 | 44.9% | 9.1% | 0.09 | 35.9% | 1.97 |
| $p \mid \lambda$ | Top-$p$ + $\lambda$-decay + $p$-reset | | | | | | | | | |
| 0.9 \| 0.9 | 41.5% | 10.3% | 0.52 | 10.3% | 3.6 | 45.4% | 6.3% | 0.57 | 9.1% | 3.9 |
| 0.9 \| 0.5 | 39.3% | 12.8% | 0.34 | 17.8% | 2.3 | 44.5% | 8.4% | 0.45 | 18.9% | 2.5 |
| $p \mid \lambda \mid \omega$ | Top-$p$ + $\lambda$-decay + $p$-reset + $\omega$-bound (*factual-nucleus sampling*) | | | | | | | | | |
| 0.9 \| 0.9 \| 0.3 | 42.1% | 10.1% | 0.55 | 7.1% | 3.8 | 46.5% | 5.6% | 0.59 | 6.4% | 4.1 |
| 0.9 \| 0.5 \| 0.3 | 41.0% | 12.2% | 0.47 | 13.0% | 2.8 | 46.0% | 7.0% | 0.51 | 12.7% | 3.0 |
| 0.9 \| 0.9 \| 0.2 | 41.7% | 9.9% | 0.52 | 8.6% | 3.6 | 45.6% | 6.2% | 0.56 | 7.6% | 4.0 |
| 0.9 \| 0.5 \| 0.2 | 39.3% | 12.8% | 0.38 | 16.1% | 2.5 | 45.2% | 7.8% | 0.42 | 16.9% | 2.7 |

Table 10: Results for factuality enhanced training. Decoding settings are formatted as: nucleus $p$ value, decay rate $\lambda$, lower-bound $\omega$

| Decoding | Factual Prompt | | | | | Nonfactual Prompt | | | | |
|---|---|---|---|---|---|---|---|---|---|---|
| $(p \mid \lambda \mid \omega)$ | $NE_{ER}\downarrow$ | $Entail_R\uparrow$ | Div. | Rep. | PPL | $NE_{ER}$ | $Entail_R$ | Div. | Rep. | PPL |
| Vanilla Pretrained LM (1.3B) | | | | | | | | | | |
| 0.9 | 52.4% | 2.9% | 0.88 | 0.2% | 10.9 | 56.8% | 2.0% | 0.89 | 0.3% | 12.0 |
| 0.9 \| 0.9 \| 0.3 | 42.1% | 10.1% | 0.55 | 7.1% | 3.8 | 46.5% | 5.6% | 0.59 | 6.4% | 4.1 |
| Factual Domain-Adaptive Training with Wikipedia (1.3B) | | | | | | | | | | |
| 0.9 | 52.5% | 2.8% | 0.85 | 0.2% | 9.73 | 55.8% | 2.2% | 0.86 | 0.1% | 10.69 |
| 0.9 \| 0.9 \| 0.3 | 42.7% | 7.1% | 0.51 | 7.2% | 3.60 | 48.2% | 4.9% | 0.56 | 6.0% | 3.95 |
| TOPICPREFIX (1.3B) | | | | | | | | | | |
| 0.9 | 34.4% | 4.2% | 0.84 | 0.3% | 8.03 | 36.2% | 2.7% | 0.85 | 0.2% | 8.61 |
| 0.9 \| 0.9 \| 0.3 | 27.6% | 8.7% | 0.43 | 8.0% | 2.60 | 30.5% | 6.1% | 0.47 | 6.9% | 2.75 |
| TOPICPREFIX + $SC_{ROOT}$ (1.3B) | | | | | | | | | | |
| 0.9 | 32.5% | 6.7% | 0.83 | 1.2% | 7.63 | 34.3% | 4.6% | 0.84 | 1.1% | 8.15 |
| 0.9 \| 0.9 \| 0.3 | 24.7% | 15.8% | 0.40 | 13.6% | 2.32 | 27.6% | 9.1% | 0.44 | 13.7% | 2.46 |
| TOPICPREFIX + $SC_{RANDOM}$ (1.3B) | | | | | | | | | | |
| 0.9 | 32.0% | 7.9% | 0.81 | 1.2% | 7.43 | 34.2% | 5.5% | 0.83 | 1.1% | 7.96 |
| 0.9 \| 0.9 \| 0.3 | 23.6% | 17.6% | 0.39 | 14.2% | 2.31 | 26.9% | 9.3% | 0.42 | 13.2% | 2.46 |
| TOPICPREFIX + $SC_{HALF}$ (1.3B) | | | | | | | | | | |
| 0.9 | 31.6% | 7.6% | 0.81 | 1.4% | 7.33 | 33.5% | 5.1% | 0.83 | 1.5% | 7.88 |
| 0.9 \| 0.9 \| 0.3 | 23.6% | 17.4% | 0.38 | 14.4% | 2.33 | 27.2% | 10.2% | 0.42 | 13.1% | 2.47 |
| TOPICPREFIX + $SC_{HALF}$ (530B) | | | | | | | | | | |
| 0.9 | 18.3% | 19.3% | 0.68 | 0.1% | 8.30 | 21.7% | 13.7% | 0.68 | 0.1% | 8.69 |
| 0.9 \| 0.9 \| 0.3 | **14.5%** | **25.5%** | 0.33 | 0.2% | 7.57 | **17.7%** | **20.0%** | 0.33 | 0.1% | 7.71 |

# G   Generation Examples (Continued)

---

**[Factual Prompt]** The ruins of the ancient Roman town of Pompeii lie near Naples

---

**[Greedy]** The city was destroyed by a volcanic eruption in 79 AD.\n\nThe ruins of the ancient Roman town of Pompeii lie near Naples. The city was destroyed by a volcanic eruption in 79 AD.\n\nPhoto: AP\n\nImage 2 of 3\n\nA man walks past a mural depicting the ancient Roman town of Pompeii in Naples, Italy.\n\nA man walks past a mural depicting the ancient Roman town of Pompeii in Naples, Italy.\n\nPhoto: AP\n\nImage 3 of 3\n\nPompeii's ruins to be restored\n\n1 / 3\n\nBack to Gallery\n\nROME

**[p=0.9]** Millions of tourists from around the world come to visit them each year. They enter through a huge gate. This gate built when Pompeii became a part of Italy again in the nineteenth century. Before that time, no one was allowed to enter the area.\n\nBut that is not the only gate through which visitors enter Pompeii.\n\nThe back of the entrance gate is very high. For a long time, no one has known why it is so tall. It seemed as if the builder simply did not finish it.\n\nOne day a

**[Ours]** Naples has a wide range of different architectural styles , from the medieval castles in the historic centre to the Art Nouveau buildings in the Vomero ärea , to the buildings of the 20th century , such as the San Carlo opera house . \nNaples is also widely known for its wealth of historical museums . \nThe Naples National Archaeological Museum is one of the city 's main museums , with one of the most extensive collections of artefacts of the Roman Empire in the world . \nIt also houses many of the antiques unearthed at Pompeii and Herculaneum , as well as some artefacts from the Greek and Renaissance periods . \nPreviously a Bourbon palace , now a museum and art gallery , the Museo di Capodimonte is another"

---

**[Factual Prompt]** David Beckham is a player for the Major League Soccer club LA Galaxy.

---

**[Greedy]** He is also a UNICEF ambassador.\n\nThe opinions expressed in this commentary are solely those of David Beckham.<|endoftext|>

**[p=0.9]** For England, he was a very successful player, playing for Man U (again and again), and captaining the team for several years, until he fell out of favor with his coach.\n\nIn 2003, after a free kick from England's David Beckham, Argentinean player and infamous cokehead Diego Simeone headbutted Beckham. According to Beckham, Simeone leered at him and said "I'm going to fuck you." Simeone claimed the players collided accidentally, but the ref gave him a red card for the incident, ...

**[Ours]** He is the first English player to win league titles in four countries : England , Spain , the United States and France . He retired in May 2013 after a 20-year career , during which he won 19 major trophies . Beckham 's professional club career began with Manchester United , where he made his first - team debut in 1992 at age 17 . With United , he won the Premier League title six times , the FA Cup twice , and the UEFA Champions League in 1999 . He then played four seasons with Real Madrid , winning the La Liga championship in his final season with the club .

---

**[Factual Prompt]** Andy Warhol has creations that are highly valuable.

---

**[Greedy]** His works are sold for millions of dollars. He is one of the most famous artists in the world.\n\nAndy Warhol was born in Pittsburgh, Pennsylvania. He was the youngest of three children. His parents were immigrants from Czechoslovakia. His father was a construction worker. His mother was a homemaker.\n\nAndy Warhol was a shy child. He was often sick. He had a nervous stomach. He was often in the hospital. He was afraid of the dark. He was afraid of the dentist. He was afraid of doctors. He was afraid of hospitals.\n\nAndy Warhol went to college. He studied art. He was a good student. He was a good artist. He was a good painter. He was a good drawer. He was a good illustrator. He was a good graphic artist...

**[p=0.9]** One of these creations is a book. During his days as an art student, Warhol used to paint advertisements that he printed on pieces of cardboard. He pasted these advertisements on a cork and this is what he called his book. For Warhol, making a book meant simply arranging his images on a page. He didn't make the necessary effort to bind the pages together to make it qualify as a book. This was the same way Warhol created many of his creations and works. He could have made every page of his work a masterpiece, but he didn't do...

**[Ours]** Warhol was an inspiration towards Cattelan 's magazine and photography compilations , such as " Permanent Food , Charley " , and " Toilet Paper " . \nIn the period just before his death , Warhol was working on "Cars", a series of paintings for Mercedes - Benz . \nA self - portrait by Andy Warhol ( 1963-64 ) , which sold in New York at the May Post - War and Contemporary evening sale in Christie 's , fetched $ 38.4 million . \nOn May 9 , 2012 , his classic painting " Double Elvis ( Ferus Type ) " sold at auction at Sotheby 's in New York for US$ 33 million . \nWith commission , the sale price totaled US$ 37,042...

---

**[Factual Prompt]** San Francisco is the location of a bridge.

**[Greedy]** The bridge is located in San Francisco.\n\nThe bridge is located in San Francisco.\n\nThe bridge is located in San Francisco.\n\nThe bridge is located in San Francisco.\n\nThe bridge is located in San Francisco.\n\nThe bridge is located in San Francisco.\n\nThe bridge is located in San Francisco.\n\nThe bridge is located in San Francisco.\n\nThe bridge is located...

**[p=0.9]** San Francisco is an island surrounded by bridges.\n\nOnce across the bridges, there is much to explore. Chinatown, for example, has interesting shops, food, and people-watching. Walk in Chinatown and you will hear many languages, see many signs in different scripts, and smell interesting things from kitchens. From Chinatown you can see the world.\n\nBut Chinatown is not what this is about. What this is about is part of San Francisco which is on the east side of the city, about three miles from Chinatown...

**[Ours]** There are seven major islands in the San Francisco Bay : Alameda , Marin , Angel , Treasure , Yerba Buena , Contra Costa and Alcatraz . Alcatraz , in the middle of the bay , is a federal penitentiary which also housed a lighthouse for many years . It is now a public museum and one of San Francisco 's major tourist attractions , attracting some 1.5 million visitors annually . Yerba Buena Island , between San Francisco and Oakland , contains the San Francisco Port of Embarkation , the Oakland Naval Supply Depot , and a Coast Guard station ...

---

**[Factual Prompt]** Michael Jordan was in "The Dream Team".

**[Greedy]** The Dream Team was the nickname of the USA men's basketball team that won the gold medal at the 1992 Olympics.\n\nThe Dream Team was the nickname of the USA men's basketball team that won the gold medal at the 1992 Olympics.\n\nThe Dream Team was the nickname of the USA men's basketball team that won the gold medal at the 1992 Olympics.\n\n

**[p=0.9]** Mickey Mouse is Disney.\n\nSophia Loren was an actress from Italy.\n\nLast but not least is Harry Potter.\n\nAll these brands immediately create a strong visual image in the mind of the target audience, and this is the same for the faith-based ministry.\n\nTo be seen as an iconic faith-based ministry the church must have a look and feel that is instantly recognizable.\n\nDo you have a recognizable logo?\n\nDoes your website and social media pages look like the brand of your church?\n\nDo you have

**[Ours]** In the summer of 1992 , Jordan led the Dream Team to the Olympic gold medal in Barcelona , scoring a game - high 26.5 points per game , and making a game - winning jump shot against Croatia . Jordan was named the tournament 's MVP , although the Olympic basketball competition was later considered to be an exhibition for NBA players , and thus NBA players were not allowed to compete in future Olympic basketball tournaments . In March 1993 , Jordan announced his retirement , citing a loss of desire to play the game . Jordan later stated that the death of his father three months earlier also shaped his decision . Jordan 's father was murdered on July 23 , 1993 , at a highway rest area in Lumberton , North Carolina , by two teenagers , Daniel Green and Larry Martin Demery , who carjacked his luxury Lexus bearing the license plate ÜNC 0023 .· His body was dumped in a South Carolina swamp and was not discovered until August 3 .

---

**[Factual Prompt]** Cara Delevingne had a role in a film.

**[Greedy]** Cara Delevingne had a role in a film. Cara Delevingne had a role in a film. Cara Delevingne had a role in a film. Cara Delevingne had a role in a film. Cara Delevingne had a role in a film. Cara Delevingne had a role in a film. Cara Delevingne had a role in a film. Cara Delevingne had a role in a film.

**[p=0.9]** She's actually one of the many faces in a sketch for a film by celebrated photographer David LaChapelle called "Lost and Found." And I'm not going to lie to you, it's kind of terrifying, but in a good way? I think.<|endoftext|>"

**[Ours]** Delevingne's first film appearance was in the 2012 film adaptation of "Anna Karenina", as Princess Sorokina, the marriage interest of Count Vronsky, alongside Keira Knightley. In August 2013, Delevingne voiced a DJ of a pop radio station in the video game "Grand Theft Auto V," which once released became the fastest - selling entertainment product in history.

**[Factual Prompt]** Doctor Who is a show on television.

**[Greedy]** It is a science fiction show. It is a show about a man who travels through time and space in a blue box. It is a show about a man who is over 900 years old. It is a show about a man who is a Time Lord. It is a show about a man who is a Time Lord who has a time machine called the TARDIS. It is a show about a man who is a Time Lord who has a time machine called the TARDIS that is bigger on the inside than it is on the outside. It is a show about a man who is a Time Lord who has a time machine called the TARDIS that is bigger on the inside than it is on the outside and that is shaped like a police box.

**[p=0.9]** The men on the show are the Doctor and the Master. The women are the companions, and they are held captive until they are rescued, or dead. When they are rescued, it's rarely by the Doctor. Even when the Doctor is in the scene, the female characters are passively waiting to be saved. Here are a few examples. \n\nSeason 5. Episode 13. "The Lodger"\n\nhttp:www.youtube.comwatch?v=bHNce0_Qlrk\n\nHere's a fun little game.

**[Ours]** Doctor Who is a British science fiction television programme produced by the BBC since 1963 . The programme depicts the adventures of a Time Lord called " the Doctor " , an extraterrestrial being , to all appearances human , from the planet Gallifrey . The Doctor explores the universe in a time - travelling space ship called the TARDIS . Its exterior appears as a blue British police box , which was a common sight in Britain in 1963 when the series first aired . Accompanied by a number of companions , the Doctor combats a variety of foes while working to save civilisations and help people in need . The show is a significant part of British popular culture , and elsewhere it has gained a cult following...

---

**[Factual Prompt]** Tiber Oil Field is operated by BP.

**[Greedy]** See also List of oil fields References Category:Oil fields in Kern County, California\nCategory:Geography of the San Joaquin Valley\nCategory:BP oil and gas fields<|endoftext|>

**[p=0.9]** The rig is located approximately off the Louisiana coast. It is a riserless subsea design, with a maximum structural height. This is the first use of this design. Operations Production on the field was expected to begin in 2011, but first oil was delayed until 2014. Oil is produced through a wellhead located on the seafloor, and it is then piped up to a floating production storage and offloading (FPSO) vessel. Oil from this vessel is then transported to shore. The project is being developed in...

**[Ours]** Tiber Oil Field is located in the Keathley Canyon block 102 of the United States sector of the Gulf of Mexico . Tiber is only one of four deepwater fields in the Gulf of Mexico that is operated by BP . The oil field was discovered in September 2009 and it is located in Keathley Canyon block 102 about southeast of Houston and south west of New Orleans . Tiber is BP 's first ultra - deepwater platform in the Gulf of...

---

**[Factual Prompt]** The Blue Lagoon was released in or before 1930.

**[Greedy]** The Blue Lagoon was released in or before 1930. The Blue Lagoon was released in 1931.\n\nThe Blue Lagoon was released in 1932.\n\nThe Blue Lagoon was released in 1933.\n\nThe Blue Lagoon...

**[p=0.9]** The film was re-released on 5 July 1949, with a higher age restriction, and three minutes of footage from the previous cut removed. The 1949 film was retitled The Island of Purity. The film was released on DVD on 6 Jan 2004...

**[Ours]** The novel has inspired several film adaptations, most notably "The Blue Lagoon" starring Brooke Shields as Emmeline and Christopher Atkins as Richard ("Dicky" in the book), in which they are two fictional English children. The story...

---

**[Factual Prompt]** Minos had a wife and he was Greek.

**[Greedy]** The Greek was a man of many words.\n\nThe Greek was a man of many words.\n\nThe Greek was a man of many words.\n\nThe Greek was a man of many words.\n\nThe Greek was a man of many words.\n\nThe Greek was a man of many words.\n\nThe Greek was a man of many words.\n\nThe Greek was a man of many words.\n\nThe...

**[p=0.9]** For this reason, this creature did not consider himself to be a Satan.\n\nEros was all about love.\n\nHe loved his wife and didn't want to hurt her.\n\nFor this reason, he also didn't believe he was an angel.\n\nLove did not originate from Hell. It's original abode was Heaven. It was only much later, when his job was downsized, he became to earn his living by showing his love through sex. This is why today, love is known to be

**[Ours]** Minos was the son of Zeus and Europa. He lived at Knossos, where he received the laws from Zeus himself. He was the founder and first king of the Minoan dynasty. He was also the judge of the underworld. In Crete , he was the founder - king of the Minoan civilization and the palace of Knossos was the centre of the Minoan civilization . On the Greek mainland , he was the founder...