# OpenReview forum: "Factuality Enhanced Language Models for Open-Ended Text Generation"
_NeurIPS.cc/2022/Conference — NeurIPS 2022 Accept_

### Official Review · Reviewer_4pX8 · 2022-07-05

**Rating:** 4
**Confidence:** 4
**Soundness:** 2 fair
**Presentation:** 2 fair
**Contribution:** 3 good

**Summary:**

This paper examines the factual incorrectness issue in text generation and proposed some decoding and training method to address these. Experimental results showed some insights regarding the effect of decoding on factuality, and the proposed methods showed improved factuality based on automatic evaluation metrics.


**Questions:**


Additional suggestions:
Providing more examples would help better understand the system output.


**Limitations:**

Authors didn't provide this in the paper, I think factuality is an important topic and worth adding some discussion for the model's limitations and societal impact of the technology.



**Strengths And Weaknesses:**

The paper addresses an important problem in text generation using large language models.

The analysis results regarding greedy and nucleus sampling decoding are intuitive. The proposed decoding method is a reasonable tradeoff between greedy and nucleus sampling, and it’s good to see that empirical results show it’s effective. Authors also provided results when only wikipedia is used, i.e., domain adaptive pertaining, there is less effect on factuality, supporting the improvement comes from their proposed approach.
Overall the paper has some interesting results, however, it leaves some questions in its current form, and is not ready for publication yet.

About the results:
Automatic metrics can give some information about the model quality. But human evaluation is an important aspect in generation evaluation and providing those results would strengthen the paper. Such results are not needed for every setup.

The proposed method lacks some clarify and justification. Specifically the factuality enhanced pretaining  is not well justified.

Unclear parts about the additional pertaining:
Authors motivated the TopicPrefix using the pronoun issues in the document. However, I don’t see why adding that prefix address the issue. Even though it’s telling the model this part is related to factual information, it can still connect with the wrong entities.
Another question is how is this prepending used during testing text generation?  Does it know for what prompts this needs to be prepended?  If so, is this a reasonable assumption?

Similarly for the sentence completion loss, it’s not well explained or justified how or why applying zero-masking for the token prediction loss helps factuality.  For different pivot it’s only used during training, during testing, nothing is needed?

---

> ### Author Response · Authors · 2022-08-02
> **Response To Reviewer 4pX8**
>
> Many thanks for your constructive comments & suggestions! We will address your comments in the following.
>
> 1, "About the results: Automatic metrics can give some information about the model quality. But human evaluation is an important aspect in generation evaluation and providing those results would strengthen the paper. Such results are not needed for every setup."
> * Many thanks for this suggestion. We have included human evaluation results in Appendix A for revision. We will include it in the main text in the final version of this paper.  We conducted human evaluation to verify if our automatic evaluation metrics correlate with human judgment, and show that they do. We focused on verifying the correlation so that the research community has an automatic metric that can be used to fairly & reliably compare between methods.
>
> 2, "Unclear parts about the additional pretraining: Authors motivated the TopicPrefix using the pronoun issues in the document. However, I don’t see why adding that prefix addresses the issue. Even though it’s telling the model this part is related to factual information, it can still connect with the wrong entities."
> * Our motivation for TopicPrefix is based on the "fact fragmentation" problem that arises from the chunking of documents at pre-training. Note that, it is common practice to chunk the documents in the corpus when training generative LMs like GPT due to GPU memory constraint. A document may use pronouns when describing the facts (e.g. He/him for Barack Obama), but if this is chunked to different sequences/segments at pretraining, this fact will be incomplete (E.g., "He is the 44th president of the United States from 2009 to 2017" can only serve as a complete fact if we know who "He" is. ). By adding the TopicPrefix (e.g. Wikipedia title), the LM could correctly associate the key topic / theme to the chunked sequence at continued pretraining, which could reduce the chance of Name Entity mix-up. See line 218 for an example.
> * We agree there is no guarantee that this will always result in the LM not mixing up entities, however, we are attempting by increasing the co-occurence of the target named-entity and its corresponding factual context/content. In qualitative study, we find that this strategy is particularly effective for biographical-style documents. The effectiveness of this strategy is also demonstrated in our quantitative study results (Table 4) where the Named-Entity Error rate of 1.3B LM drops from 52.4% to  34.4% even with simple top-p sampling.
>
> 3, "Another question is how is this prepending used during testing text generation? Does it know for what prompts this needs to be prepended? If so, is this a reasonable assumption?"
> * We tested with a setting where TopicPrefix is prepended in the same format as how it was trained. In practice, we can obtain keywords that can serve as TopicPrefix through approaches such as applying named entity recognition, using open-information-extraction (OpenIE) modules [1,2], or adopting "search query generation" that are used in works such as [3]. In particular, the NE extraction can work well if the prompts are about some named entities.
> [1] Open Language Learning for Information Extraction
> [2] https://stanfordnlp.github.io/CoreNLP/openie.html
> [3] Language Models that Seek for Knowledge: Modular Search & Generation for Dialogue and Prompt Completion.
>
> 4, "Similarly for the sentence completion loss, it’s not well explained or justified how or why applying zero-masking for the token prediction loss helps factuality."
> * The motivation of introducing sentence completion loss is that the latter part of sentence is more critical for factuality during text generation. For example, the LM generates a non-factual sentence “Samuel Witwer’s father is a Lutheran minister”. This sentence is nonfactual because LM failed to generate factual content after “Samuel Witwer’s father is”. Note that, LM is pre-trained to predict every subword token within a sentence given the limited model capacity. As a result, training LM to only predict the tokens after the pivot will encourage LM focusing on the prediction of the latter part of the sentence, especially given the limited model capacity. This can help factuality, and we show its effectiveness from our experimental results in Table 4. We will further clarify this in the final version of the paper.
>
> 5, "For different pivots it’s only used during training. During testing, nothing is needed?"
> * No pivots are needed during testing/inference time. "Pivot" is only required for training sentence completion loss. We will clarify this in the paper revision.

---

> ### Author Response · Authors · 2022-08-02
> **Response To Reviewer 4pX8 (continued)**
>
> 6. "Additional suggestions: Providing more examples would help better understand the system output."
> * We appreciate your nice suggestion. In the revision, we provide more examples in Appendix C.  There are overall 22 examples in the manuscript now. Please note that we have also included some generation examples from "nonfactual prompts" (pg "18" in appendix), and show that our generations are more robust to nonfactual prompts. We will add more examples to the main text in the final version.
>
> 7. "Limitations: Authors didn't provide this in the paper, I think factuality is an important topic and worth adding some discussion for the model's limitations and societal impact of the technology."
> * Thanks for pointing this out. We added the following discussion of limitation/potential negative impact in the Appendix G of our updated paper:
>
>     * Fine-tuning large models will have some negative carbon footprint. However, our task itself (trying to improve factuality) will bring more overall benefit to the community and society, by allowing the language models to generate less fake information and be safer.
>     * To the best of our knowledge, Wikipedia has no personal information or any inappropriate content (problematic discrimination towards particular demographic groups, NSFW contents, hate speech, etc).

---

> ### Author Response · Authors · 2022-08-08
> **Looking forward to hearing your feedback**
>
> Dear Reviewer 4pX8,
>
> We want to thank you again for your insightful comments and valuable suggestions. They are really helpful to improve the quality of our paper.
>
> We follow your suggestions and revise the papers accordingly. Specifically, we did the following:
> 1. included human evaluation results in Appendix A
> 2. Included more illustrative examples in Appendix C
> 3. Included discussion of potential negative impact in Appendix G
>
> Also, we have provided clarification regarding the factuality enhanced pre-training. We are wondering if our responses have addressed your concerns. Please let us know if you have any further questions.
>
> Thanks a lot!

---

> ### Author Response · Authors · 2022-08-08
> **Human evaluation results**
>
> Dear Reviewer,
>
> Per your suggestion, we have provided the human evaluation results in the paper revision (Appendix A).  For your convenience, we also provide the results directly in this comment section.
>
> To ensure the validity of our factuality metrics ($\text{NE}_{\text{Er}}$ and $\text{Entail}$), we collect human annotations for 200 randomly chosen generations to evaluate the correlation between our automatic factuality metrics with human judgment. The below table shows the Pearson correlation coefficients between human factuality annotations and our automatic metrics. p-values for all results are less than 0.01.
>
> | Annotation  |  $\text{Entail}$$\uparrow$  | $\text{NE}_{\text{Er}}$$\downarrow$  |
> |-----------------|:---------------------------:|:-------------------:|
> | Expert          |             0.81            |        -0.77        |
> | Majority Voting |             0.47            |        -0.46        |
>
> Our results show that there are strong correlations between human judgment of factuality and the proposed automatic metric $\text{NE}_{\text{Er}}$ and $\text{Entail}$, especially with the expert annotations.
>
> Some details about the human evaluation:
> - The annotators are asked to fact-check the LM continuations against Wikipedia and assign a factuality label (1 = Factual : can find supporting Wikipedia evidence. 0 = Non-factual : cannot find supporting Wikipedia evidence).
> - We provide two types of annotations — expert annotation and crowd-sourced annotation. Expert annotation is included because fact-checking annotation is a challenging and time-consuming task which requires careful reasoning over multiple pieces of evidence.
>
> More details can be found in Appendix A.
>
> Thank you!

---

### Official Review · Reviewer_TZp5 · 2022-07-11

**Rating:** 5
**Confidence:** 3
**Soundness:** 2 fair
**Presentation:** 3 good
**Contribution:** 2 fair

**Summary:**

In this paper, the authors highlight the factuality problem in the generation of language modeling. And they proposes a two-fold solution to improve factuality. The first part includes several sampling adaptations to avoid unnecessary logits sacrifice according to the growing sequence length. The second part includes a topic prefix method and sentence completion as the additional loss. The results show that their methods reduce the entity and entailment error in a factual-based perspective.

**Questions:**

It would be helpful to elaborate on line 303.

**Limitations:**



**Strengths And Weaknesses:**

Strength
- The entity error and entail error measurements are reasonable criteria for factuality measurement.
- The observation of randomness in nucleus sampling adds value to better understand the factuality in language generation.

Weaknesses
- It seems that the entity ratio is not the whole world of factuality, organization and representation of sentences, is important as well apart from the entity mentions (e.g. negations). It would better if the experiments are conducted on some factual intensive tasks or having some human evaluations to help better interpret the results.

---

> ### Author Response · Authors · 2022-08-02
> **Response To Reviewer TZp5**
>
> Many thanks for your review! It is really helpful to improve the quality of our paper. We will address your concern & question in the following.
>
> 1, “It seems that the entity ratio is not the whole world of factuality, organization and representation of sentences is important as well apart from the entity mentions (e.g. negations). It would be better if the experiments are conducted on some factual intensive tasks or having some human evaluations to help better interpret the results."
> * Yes, the proposed entailment-based metric using natural language inference model is actually designed to capture the semantic/directional relation between our generation and the ground truth knowledge source. This will consider more complex representation of sentences beyond merely looking at named entities.
>
> * Many thanks for your nice suggestion. We have included human evaluation results in the updated Appendix A (and if accepted, we will include it in the main manuscript in the additional page). We conducted human evaluation to verify if our automatic evaluation metrics correlate with human judgment, and show that they do. We focused on verifying the correlation so that the community has an automatic metric that can be used to easily & reliably compare between methods. We also included the human annotation files (both expert and crowdsourced annotations) in the updated supplementary.zip file for interested reviewers to refer to.
>
> 2, "It would be helpful to elaborate on line 303."
> * To elaborate on line 303 content ("Surprisingly, according to our results, the simplest SC_HALF performs on par with the complex ones."): We experimented with three different designs in deciding on the pivot position for the proposed sentence completion loss. We initially speculated that the complex SC_ROOT pivot (i.e., apply dependency parsing to a sentence to identify the ROOT node/head of that sentence) will perform better than SC_HALF. However, our results showed that the simple strategy (SC_HALF), which always sets the pivot at the half of a sentence, works as well as the complex strategy.

---

> ### Author Response · Authors · 2022-08-09
> **Looking forward to hearing whether our response have addressed your concerns**
>
> Dear Reviewer,
>
> Many thanks again for your review!
>
> We hope our response could address your major concerns.  In particular, per your nice suggestion, we have included the human evaluation results in Appendix A of the paper revision. For your convenience, we also introduce the results here.
>
> We collect human annotations for 200 randomly chosen generations to evaluate the correlation of our automatic factuality metrics ($\text{NE}_{\text{Er}}$ and $\text{Entail}$) with human judgment. The below table shows the Pearson correlation coefficients between human factuality annotations and our automatic metrics. p-values for all results are less than 0.01.
>
> | Annotation  |  $\text{Entail}$$\uparrow$  | $\text{NE}_{\text{Er}}$$\downarrow$  |
> |-----------------|:---------------------------:|:-------------------:|
> | Expert          |             0.81            |        -0.77        |
> | Majority Voting |             0.47            |        -0.46        |
>
> Our results show that there are strong correlations between human judgment of factuality and the proposed automatic metric $\text{NE}_{\text{Er}}$ and $\text{Entail}$, especially with the expert annotations.
>
> Details about the human evaluation:
> - The annotators are asked to fact-check the LM continuations against Wikipedia and assign a factuality label (1 = Factual : can find supporting Wikipedia evidence. 0 = Non-factual : cannot find supporting Wikipedia evidence).
> - We provide two types of annotations — expert annotation and crowd-sourced annotation. Expert annotation is included because fact-checking annotation is a challenging and time-consuming task which requires careful reasoning over multiple pieces of evidence.
>
> More details can be found in Appendix A.
>
> In the remaining time of the author-reviewer discussion period, we hope to have the opportunities to discuss with you about our work, and answer additional questions you may have. Thank you so much again for the feedback!

---

### Official Review · Reviewer_wMPr · 2022-07-12

**Rating:** 7
**Confidence:** 4
**Soundness:** 3 good
**Presentation:** 3 good
**Contribution:** 3 good

**Summary:**

This paper addresses the tendency of open generation systems to produce factually incorrect sequences. In order to quantify and measure factual correctness of text generative models, a dataset comprising factual and wrong prompts is created. This dataset is created from a fact checking dataset FEVER that has true and fake documents with their corresponding Wikipedia sources. The proposed dataset uses FEVER to obtain the linked Wikipedia article for each prompt and defines metrics mainly focused on named entities and entailment of generation by the corresponding sources. This paper finds that larger LMs are more factually correct, but in general tend to mix and match the named entities with wrong facts or fabricate entirely new named entities, especially with nucleus sampling. The authors propose to vary the nucleus parameter during nucleus sampling as the decoding proceeds so that there is more randomness at the start of a sentence and less randomness as it ends, so that nucleus sampling generates factually correct sequences.

The paper also proposes fine tuning objectives to improve factual correctness. One of the strategies is to prepend sequences with topic titles (obtained via Wikipedia titles), and the other approach is to use the latter part of sentences for loss computation with the idea being that facts are generated toward the end which have to be consistent with the information generated in prior steps. Empirical analysis is performed to test the proposed approaches.

**Questions:**

Please find my comments in the section above.

**Strengths And Weaknesses:**

The identified problem is important and the associated dataset and metrics will be a useful resource for factuality assessment of generative models. That said, the scope of the dataset is limited to Wikipedia articles and hence the metrics might be biased toward Wikipedia-like language. Extending to other kinds of knowledge bases does not seem very straightforward as this approach depends on a pre-exisitng dataset called FEVER which in turn heavily depends on Wikipedia.

The findings that large LMs are more factual is interesting and the proposed fixes also show that the larger LMs show greater improvement.

Nucleus sampling is compared with greedy decoding which is a bit unfair. It should be compared with other sampling approaches like ancestral sampling, sampling with a range of temperatures etc. Also, the proposed fix seems ad-hoc as it encourages deterministic output toward latter parts of the generation. However, the prompt itself sets up some background so following the proposed motivation, there should be less stochasticity in the generations immediately following the prompt as well. It should be noticed that barring diversity, greedy decoding is superior on other metrics. Comparison with temperature based sampling will help -- especially if the same scheme is applied to the evolution of temperature as is applied to the nucleus parameter. This would shed more light on the effect of reducing stochasticity gradually while generating.

The proposed training fixes are simple heuristics as well but the results do seem to suggest that they are very effective at improving factuality generation. However, it should be noticed that fine tuning is done on Wikipedia as well which favors the proposed dataset and metrics. Experiments with fine tuning on factual data from sources other than Wikipedia would be helpful. Also, these approaches do require finetuning of a large model though which can be expensive.

Overall, the paper has important contributions but they come with some caveats.

---

> ### Author Response · Authors · 2022-08-02
> **Response To Reviewer wMPr**
>
> Many thanks for your detailed comments! They are really helpful for improving our paper. We address your comments below:
>
> 1, "The scope of the dataset is limited to Wikipedia articles and hence the metrics might be biased toward Wikipedia-like language. Extending to other kinds of knowledge bases does not seem very straightforward as this approach depends on a pre-existing dataset called FEVER which in turn heavily depends on Wikipedia."
> * Our metric is designed independently to the type of knowledge source being used, so it won't be biased towards the Wikipedia-like language. The metric is based on linguistic models such as textual entailment models and Named-Entity-Recognition models that are general purpose NLP techniques.
> * We also would like to point out that our test prompts were constructed from FEVER dataset, because it allowed us to obtain high quality factual prompts to test the models without collecting our own human annotation. However, if more test prompts are constructed about other kinds of free-form knowledge base, our metric can easily be used to evaluate it as long as we provide access to that knowledge base for retrieval.
>
> 2, "Nucleus sampling is compared with greedy decoding which is a bit unfair. It should be compared with other sampling approaches like ancestral sampling, sampling with a range of temperatures etc. Also, the proposed fix seems ad-hoc as it encourages deterministic output toward latter parts of the generation. However, the prompt itself sets up some background so following the proposed motivation, there should be less stochasticity in the generations immediately following the prompt as well. It should be noticed that barring diversity, greedy decoding is superior on other metrics. Comparison with temperature based sampling will help -- especially if the same scheme is applied to the evolution of temperature as is applied to the nucleus parameter. This would shed more light on the effect of reducing stochasticity gradually while generating.”
> * Thanks for suggesting sampling with a range of temperatures. We used the nucleus sampling as the baseline, because it is very popular in open-ended text generation. The main contribution of this work is to show that the stochasticity from sampling can lead to factual errors, and thus, controlling for it helps to improve factuality. Tuning the temperature in ancestral sampling has a similar effect as tuning the nucleus p in nucleus sampling (i.e., changing the strength of sampling), so we believe dynamic evolution of temperature through time will have a similar effect to dynamic evolution of nucleus parameters.
> * For a fair and thorough comparison between our "factual-nucleus sampling with evolving stochasticity" and "standard nucleus sampling with static stochasticity", we also compared it with a range of nucleus values. Please refer to Figure 2 in Appendix B in the updated manuscript. We observe that the family of factual-nucleus sampling algorithms achieves clearly better trade-offs between factuality and diversity/repetition.
> * Except for bland and less diverse generations, another major issue of greedy decoding (also beam search)  for open-ended text generation is the serious repetition. The effective heuristics to reduce repetition in machine translation (e.g., length normalization, coverage penalty given the source sentence [1]) are not available for open-ended generation. As a result, greedy decoding is rarely used for open-ended generation. See [2] for more detailed analysis.
>
> [1] Google's neural machine translation system: Bridging the gap between human and machine translation.
>
> [2] The Curious Case of Neural Text Degeneration.
>
>
> 3, "Experiments with fine tuning on factual data from sources other than Wikipedia would be helpful. Also, these approaches do require finetuning of a large model though which can be expensive."
> * Thanks for your suggestion. We mainly used Wikipedia because it is a publicly accessible data that is commonly used in factuality related works such as grounded dialogue generations and retrieval augmented generations. We think experiments with factual data from other domains would be valuable for future work.
> * Indeed, fine-tuning a large LM can be expensive. Depending on the resource availability and the needs, researchers/practitioners can choose between our decoding-time solution and fine-tuning based solution; factual-nucleus sampling has the same computational cost as regular top-p sampling while providing a good improvement of factuality without any fine-tuning.
> * Exploring more efficient decoding-time methods without fine-tuning would be interesting for future study. We tried to apply DExperts [3], a state-of-the-art method for controlled text generation, but observing negative results so far.
>
> [3] DExperts: Decoding-Time Controlled Text Generation with Experts and Anti-Experts

---

### Official Review · Reviewer_2ne3 · 2022-07-22

**Rating:** 6
**Confidence:** 4
**Soundness:** 3 good
**Presentation:** 4 excellent
**Contribution:** 3 good

**Summary:**

This papers proposes FACTUALITYPROMPTS, a new benchmark for open-ended text generation factuality evaluation. The dataset used for the benchmark is selected from FEVER. Based on the document-level and sentence-level factuality, the authors adapt two evaluation metrics for factuality evaluation focusing on named entities hallucination and textual entailment. Using the proposed benchmarks, the authors test language models with different sizes and also compare greedy decoding and top-p sampling. Informed by their observation that greedy decoding leads to more factually consistent generation and that tokens appearing later in the sentences are more important, the authors proposed a modified top-p decoding with decaying p within sentences (factual-nucleus sampling). Lastly, the authors presents a modified training method to enhance factuality, which mask off training signals from first few tokens in the sentence (tokens to the left of a “sentence pivot” token).

**Questions:**

Please see weaknesses 1 - 3.

**Limitations:**

The authors should address if the proposed benchmark and the trained models may have any unfair biases or other potential negative impact.

**Strengths And Weaknesses:**

Strengths:

1. The authors conduct extensive experiments to quantify the level of factuality in text generation, and how their proposed inference and training method improve factuality.

2. The paper is innovative as most previous works target factuality for specific downstream tasks (such as text summarization) but this work focus on general purpose open-domain text generation

3. The paper is well-written.


Weaknesses:

1. From Section 6, I am not sure if the proposed training method "topic-prefix" and "sentence completion loss" individually could bring significant improvement in smaller-size language models.

2. The proposed factuality-based training requires “topic prefix” which may not be available in other domains like "arXiv papers, medical reports". It would be helpful if the authors could address that limitation / discuss how to obtain such topic prefix from domains outside of wikipedia.

3. It is not clear how the "range of language models" with different sizes are trained and tested, e.g. the data (and domains) used to pre-train these language models. Can we decouple the factuality from the pre-training data from factuality that the model learns from the authors' fine-tuning data?

---

> ### Author Response · Authors · 2022-08-02
> **Response To Reviewer 2ne3 (Part 1)**
>
> Thank you so much for your review. The comments are helpful to further improve our paper. We will address your comments in the following.
>
> **1**, "From Section 6, I am not sure if the proposed training methods "topic-prefix" and "sentence completion loss" individually could bring significant improvement in smaller-size language models."
> * In Table 4, we observe that the improvement is more significant in a smaller LM (1.3B) than a bigger LM (530B). For example, the $\text{NE}_{\text{Er}}$ of the 1.3B model (top-p 0.9) is reduced from 52.4% to 31.6% ($\downarrow$ 20.8%) with the combination of "topic-prefix" and "sentence completion loss using pivot=HALF".
>
> * In contrast, the $\text{NE}_{\text{Er}}$ of the 530B is reduced from 33.3% to 18.3% ($\downarrow$ 15%). This is not surprising as smaller-size LMs perform worse, thus there is more room for improvement. This trend holds across different LM sizes (see Table 2).
>
> * We provided "topic-prefix"-only results in Table 4 at submission. The results showed that the proposed "topic-prefix" is very effective individually, especially for reducing $\text{NE}_{\text{Er}}$ -- e.g., 52.4% to 34.4% with regular top-p sampling.
>
> * For “sentence completion loss”-only, we have conducted smaller scale experiments (w/ 3k factual prompts for test, rather than 8k factual prompts) due to limited amount of time at rebuttal. We will add finalized “sentence completion loss”-only results in the final version per your suggestion. The current results suggest: the sentence completion loss is indifferent to having the standard factual-domain adaptive training (i.e., very negligible difference in factuality). However, when used together with "topic-prefix", it further improves $\text{NE}_{\text{Er}}$ and provides a significant boost for $\text{Entail}$ metric. Below is the ablation experiments for sentence completion loss:
> | Model Choice             | Decoding   | $\text{NE}_{\text{Er}}$$\downarrow$ | $\text{Entail}$$\uparrow$ |
> |--------------------------|------------|----------------------|------------------|
> | Default Wiki FT baseline | 0.9        | 55.92%               | 2.58%            |
> |                          | 0.9 \| 0.9 | 45.78%               | 7.12%            |
> | $SC_{\text{ROOT}}$                | 0.9        | 55.81%               | 2.36%            |
> |                          | 0.9 \| 0.9 | 44.80%               | 6.39%            |
> | $SC_{\text{ROOT}}$ + TopicPrefix  | 0.9        | 35.15%               | 6.67%            |
> |                          | 0.9 \| 0.9 | 26.04%               | 15.67%           |
> | $SC_{\text{HALF}}$                | 0.9        | 56.16%               | 2.08%            |
> |                          | 0.9 \| 0.9 | 45.13%               | 6.64%            |
> | $SC_{\text{HALF}}$ + TopicPrefix  | 0.9        | 34.18%               | 7.63%            |
> |                          | 0.9 \| 0.9 | 25.15%               | 18.58%           |
>
>
>
> **2**, "The proposed factuality-based training requires “topic prefix” which may not be available in other domains like "arXiv papers, medical reports". It would be helpful if the authors could address that limitation / discuss how to obtain such topic prefix from domains outside of wikipedia."
> * This is a good question. We try to address this question below and will include the discussion in our final manuscript. Potential solutions could be to: i) use the full title of the arXiv or medical reports, ii) use the keyword extracted from the titles using techniques such as Named-Entity-Recognition or Open-Information-Extraction (OpenIE), iii) do topic modeling on that domain and use the representative topic keywords for each arXiv/medical reports. Exploring the solutions for other domains can be interesting for future research.
>
> **3**, "It is not clear how the "range of language models" with different sizes are trained and tested, e.g. the data (and domains) used to pre-train these language models. Can we decouple the factuality from the pre-training data from factuality that the model learns from the authors' fine-tuning data? "
> * This is a good question for clarification. Indeed, all LMs with different sizes are pre-trained on the same corpus, i.e., a variant of the pile dataset, which contains both factual and non-factual information. Then, all pre-trained LMs are fine-tuned on the same Wikipedia corpus with topic-prefix in factual domain, although we showed that direct fine-tuning does not work (see Factual Domain-Adaptive Training with Wikipedia in Table 4).  We will clarify this in the paper revision.

---

> ### Author Response · Authors · 2022-08-02
> **Response To Reviewer 2ne3 (Part 2)**
>
> **4**, "The authors should address if the proposed benchmark and the trained models may have any unfair biases or other potential negative impact."
> * Thanks for pointing this out. We added the following discussion of limitation/potential negative impact in the Appendix G of our paper:
>
>     * Fine-tuning large models will have some negative carbon footprint. However, our task itself (trying to improve factuality) will bring more overall benefit to the community and society, by allowing the language models to generate less fake information and be safer.
>     * To the best of our knowledge, Wikipedia has no personal information or any inappropriate content (problematic discrimination towards particular demographic groups, NSFW contents, hate speech, etc).

---

### Public Comment · ~Nayeon_Lee1 · 2023-03-02
**Release of our evaluation code base**

Our evaluation code is publicly available on Github: https://github.com/nayeon7lee/FactualityPrompt

---

### Meta-Review · Area_Chair_hrVd · 2022-08-27

**Recommendation:** Accept
**Confidence:** Certain

**Metareview:**

This paper proposed a new dataset and a new benchmark for factuality of open-ended text generation.  Based on an analysis on the factuality of language models with different sizes using this benchmark, this paper proposed a modified top-k sampling strategy and a modified training method to improve the factuality of text generation.  The work is solid.  Three of four reviewers give positive ratings while reviewer 4pXB negative.  The author(s) addressed the concerns adequately but the reviewer 4pXB did not reply.

**Award:**

No

---

### Decision · Program_Chairs · 2022-09-14

Accept